# SKI to go Faster: Accelerating Toeplitz Neural Networks via Asymmetric Kernels

## Abstract

Toeplitz Neural Networks (TNNs) [1] are a recent impressive sequence model requiring $O(n \log n)$ computational complexity and $O(n)$ relative positional encoder (RPE) multi-layer perceptron (MLP) and decay bias calls. We aim to reduce both. We first note that the RPE is a non symmetric positive definite kernel and the Toeplitz matrices are pseudo-Gram matrices. Further 1) the learned kernels display spiky behavior near the main diagonals with otherwise smooth behavior; 2) the RPE MLP is slow. For bidirectional models, this motivates a sparse plus low-rank Toeplitz matrix decomposition. For the sparse component's action, we do a small 1D convolution. For the low rank component, we replace the RPE MLP with linear interpolation and use Structured Kernel Interpolation (SKI) [2] for $O(n)$ complexity. For causal models, "fast" causal masking [3] negates SKI's benefits. Working in frequency domain, we avoid an explicit decay bias. To enforce causality, we represent the kernel via the real part of its frequency response using the RPE and compute the imaginary part via a Hilbert transform. This maintains $O(n \log n)$ complexity but achieves an absolute speedup. Modeling the frequency response directly is also competitive for bidirectional training, using one fewer FFT. We improve on speed and sometimes score on the Long Range Arena (LRA) [4].

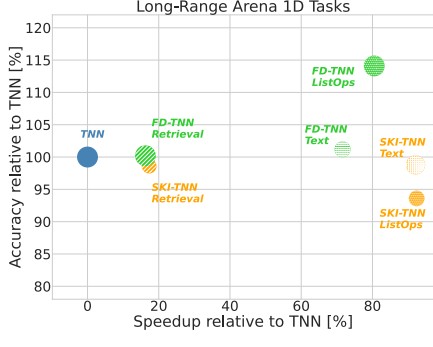
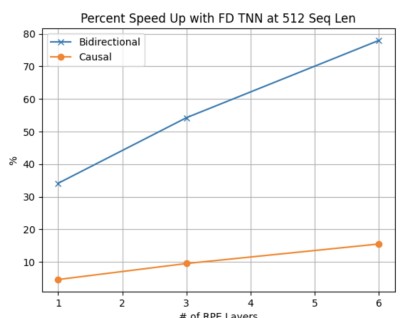

(a) Long Range Arena (LRA).  (b) Pre-training speedups using Fourier Domain.

Figure 1: (a) In LRA, our approaches, SKI and FD-TNN are faster than TNNs for 1d tasks with strong LRA scores. Bubble sizes denote training model memory. (b) Our approach, FD-TNN, achieves substantial speed ups in iterations/sec for pre-training both causal and bidirectional models. Note that we do not include SKI-TNN in this plot as it does not use an MLP based RPE.

Submitted to 37th Conference on Neural Information Processing Systems (NeurIPS 2023). Do not distribute.

# 1  Introduction

Sequence modeling is important in natural language processing, where sentences are represented as a sequence of tokens. Successful sequence modeling typically involves token and channel mixing. Token mixing combines representations of different sequence parts, while channel mixing combines the information across different dimensions of embedding vectors used to encode tokens. Transformers [5] are arguably the most successful technique for sequence modeling, and variants including [6, 7] have achieved state of the art performance on natural language tasks. They use self-attention for token mixing and feedforward networks for channel mixing.

Recently, [1] proposed Toeplitz Neural Networks (TNN) using Toeplitz matrices for token mixing. They use a learned neural similarity function, the Relative Positional Encoder (RPE), to form the Toeplitz matrices. Toeplitz matrix vector multiplication can be performed with sub-quadratic complexity using the Fast Fourier Transform (FFT), giving the TNN token mixing layer a total $O(dn \log n)$ computational complexity, where $d$ is the embedding dimension and $n$ is the sequence length. This achieved state of the art predictive performance and nearly state of the art speed for the long range arena (LRA) benchmark [4]. They also showed strong performance pre-training wikitext-103 [8] and on the GLUE benchmark[9]. Despite strong empirical speed performance, TNNs have two fundamental efficiency limitations: 1) super-linear computational complexity 2) many calls to the RPE: for each layer, one call per relative position.

In this paper, we interpret the RPE as a non-SPD kernel and note 1) the learned kernels are discontinuous near the main diagonals but otherwise smooth globally; 2) the ReLU RPE learns 1D piecewise linear functions: an MLP is slower than necessary. For bidirectional models, this motivates a sparse plus low-rank decomposition. We apply the sparse component's action via a small 1D convolution. For the low rank component, we replace the RPE MLP with linear interpolation at a set of inducing points and an asymmetric extension of Structured Kernel Interpolation (SKI) [2] for $O(n)$ complexity. Further, using an inverse time warp, we can extrapolate beyond sequence lengths observed during training. For causal models, even "fast" causal masking [3] negates the speed and memory benefits from SKI. Thus, we instead represent the real part of the kernel's frequency response using the RPE MLP, and evaluate the RPE with finer frequency resolution to extrapolate to longer sequence lengths in the time domain. From the real part, we compute the imaginary part via a Hilbert transform during the forward pass to enforce causality. In the bidirectional setting, we remove the causality constraint and represent the complex frequency response of the kernel with the RPE MLP. Levels of smoothness in frequency response imply decay rates in the time domain: thus we model the decay bias implicitly. This maintains $O(n \log n)$ complexity but achieves an absolute speedup. Further, it often leads to better predictive performance on LRA tasks.

This paper has three primary contributions: 1) a TNN sparse plus low rank decomposition, extending SKI to TNNs for the low rank part. We replace the RPE MLP with linear interpolation and apply inverse time warping to efficiently train bidirectional TNNs. We provide rigorous error analysis for our asymmetric SKI application; 2) alternatively, for both causal and bidirectional models, we work directly in the frequency domain and use the Hilbert transform to enforce causality in the autoregressive setting. We prove that different activation choices for an MLP modeling the discrete time Fourier transform (DTFT) lead to different decay rates in the original kernel. 3) Empirical results: we demonstrate that our approaches show dramatically improved computational efficiency, setting a new speed state of the art on LRA [10] on the 1d tasks, with strong LRA score. In section 2 we describe related work. In section 3 we propose our new modeling approaches. In 4 we state several theoretical results regarding our modeling approaches. In 5 we extend the empirical results of [1], showing our speed gains with minimal prediction deterioration. We conclude in section 6

# 2  Related

The most related papers use Toeplitz matrices for sequence modeling [1, 11, 12]. We build off of [1] and introduce several techniques to improve on their speed results. [11] took a similar approach, but applied Toeplitz matrices to self-attention rather than departing from it. [12] is also similar, using alternating Toeplitz and diagonal matrices as a replacement for self-attention within a Transformer. While we focus on the setting of [1] as it was released first, our approach is applicable to [12].

Also related are kernel based xFormers, particularly those using the Nyström method [13, 14]. The most related work is [15], which adapts a matrix Nyström method for asymmetric matrices [16] to self-attention. We instead adapt this along with SKI [2] to Toeplitz matrices. [17] extends [15] by embedding the self-attention matrix into a larger PSD kernel matrix and approximating the larger matrix instead. Their final approximate matrix has lower spectral error compared to [15] and higher average validation accuracy on LRA [4]. However, their method is slightly slower. Also somewhat related are random feature self-attention approximations[18, 19]. These extend [20], but use different random features that better approximate self-attention than random Fourier or binning features.

Sparse transformers are also relevant. [21] proposed using strided and fixed patterns. [22] alternated between sparse locally banded and dense attention. Finally, [23] proposed combining random attention, window attention and global attention. Our use of a short convolutional filter is most similar to window attention. The space of efficient transformers is huge and there are many models that we haven't covered that may be relevant. [10] provides an excellent survey.

Other successful long sequence approaches include state space models [24, 25, 26], long convolution [27, 28], adding moving averages to gated attention [29] and more [30].

# 3 Modeling Approach

We review Toeplitz neural networks (TNNs) in section 3.1. We next speed up the TNN's Toeplitz neural operator (TNO). We discuss using Nyström and SKI approaches to bidirectional training in 3.2. We discuss frequency based approaches, particularly for causal training in 3.3.

## 3.1 Preliminaries: Toeplitz matrices and Toeplitz Neural Networks

TNNs [1] replace self-attention, which computes the action of self-attention matrices that encode the similarity between both observation values and absolute positions, with the action of Toeplitz matrices that encode similarity only based on *relative* positions. Toeplitz matrices have, for each diagonal, the same entries from left to right. That is, $\mathbf{T}_{ij} = t_{i-j}, \mathbf{T} \in \mathbb{R}^{n \times n}$. Unlike self-attention matrices, which require $O(n^2)$ memory, a Toeplitz matrix has $2n - 1$ unique elements and requires $O(n)$ memory. Due to close connections with discrete-time convolution, $\mathbf{Tx}$ can be computed in $O(n \log n)$ time by embedding $\mathbf{T}$ in a circulant matrix and applying FFT.

A TNN [1] has multiple sequence modeling blocks, which we show in Figure 3 in Appendix A. Each block has a Gated Toeplitz Unit (GTU), which does both token and channel mixing, followed by a Gated Linear Unit (GLU) [31], which does channel mixing. The core of the GTU is the Toeplitz Neural Operator (TNO), which does token mixing and is the part of the architecture that we modify.

We now describe the TNO, shown in Figure 3b of Appendix A. Given a sequence $\mathbf{X} \in \mathbb{R}^{n \times d}$ of length $n$ and dimension $d$ in discrete time, there are $2n - 1$ unique relative positions/times $i - j$ for $i, j = 1, \ldots, n$. An RPE $: \mathbb{Z} \to \mathbb{R}^d$ neural network maps each relative position to a $d$-dimensional embedding. These embeddings are used to construct Toeplitz matrices $\mathbf{T}^l$ for $l = 1, \ldots, d$ using

$$\mathbf{T}_{ij}^l = \lambda^{|i-j|}\mathrm{RPE}_l(i - j).$$

$\mathrm{RPE}_l(i - j)$ is a learned similarity between positions for dimension $l$, while $\lambda^{|i-j|}$ with $\lambda \in (0, 1)$ is an exponential decay bias penalizing far away tokens to be dissimilar. We can interpret $\mathbf{T}_{ij}^l$ as evaluating a stationary non-SPD kernel $k_l(i - j) = \lambda^{|i-j|}\mathrm{RPE}_l(i - j)$. Thus $\mathbf{T}^l$ can be interpreted as a pseudo or generalized Gram matrix. Letting $\mathbf{x}^l$ be the $l$th column of $\mathbf{X}$, the TNO outputs

$$\mathrm{TNO}(\mathbf{X}) = (\mathbf{T}^1\mathbf{x}^1 \ldots \mathbf{T}^d\mathbf{x}^d) \in \mathbb{R}^{n \times d}$$

where each $\mathbf{T}^l\mathbf{x}^l$ is computed via the FFT as described above.

The main costs are the RPE's MLP, the FFT, and the decay bias. We aim to eliminate the MLP and decay bias when possible. In the bidirectional setting, we use SKI to apply the FFT using a much smaller Toeplitz matrix. In a separate model we learn the RPE's frequency response directly. In the bidirectional setting, this allows us to both avoid explicitly modeling the decay bias and use one fewer FFT. In the causal setting, it allows us to avoid explicitly modeling the decay bias.

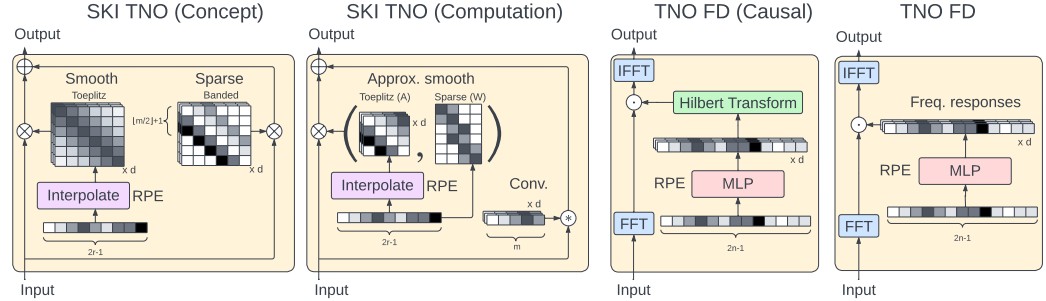

| (a) SKI-TNO. | (b) Fast implementation of SKI-TNO. | (c) FD-TNO causal. | (d) FD-TNO bidirectional. |

Figure 2: Our SKI-TNO and FD-TNO modifications: (a) We decompose Toeplitz matrices into sums of sparse + smooth components. Additionally, we use interpolation instead of an MLP to learn the RPE. (b) We use a 1D convolution to apply the sparse component and SKI as a low-rank approximation to the smooth component. (c) For the causal case, we use frequency domain RPE with a Hilbert Transform to enforce causality. (d) Our FD-TNO also is competitive in the bidirectional case, with one fewer FFT than TNO.

## 3.2 SKI Based Approaches for Bidirectional Training

For a given Toeplitz matrix $\mathbf{T}$, we assume it admits a decomposition that we can approximate with a sparse+low-rank representation, $\mathbf{T} = \mathbf{T}_{\text{sparse}} + \mathbf{T}_{\text{smooth}} \approx \mathbf{T}_{\text{sparse}} + \mathbf{T}_{\text{low}}$. Our bidirectional training thus consists of three primary components. The first, the sparse component $\mathbf{T}_{\text{sparse}}$ is straightforward. Applying the action $\mathbf{T}_{\text{sparse}}\mathbf{x}$ of $\mathbf{T}_{\text{sparse}} \in \mathbb{R}^{n \times n}$ with $m$ non-zero diagonals is equivalent to applying a 1D convolution layer with filter size $m$. We then discuss our asymmetric SKI for $\mathbf{T}_{\text{low}}$ in section 3.2.1. Finally, we discuss how we handle sequence lengths not observed in training for $\mathbf{T}_{\text{low}}$ via an inverse time warp in section 3.2.2. Algorithm 1 summarizes our TNO based on these techniques.

---

**Algorithm 1** Sparse Plus Low Rank Bidirectional TNO with Asymmetric SKI

---

    **Given** sequence $\mathbf{X} \in \mathbb{R}^{n \times d}$ with columns $\mathbf{x}^l$
    **Hyperparameters** rank $r \ll n$, sparse filter size $m$, interpolation degree $N$, decay parameter $\lambda$
    **Compute** inducing points $p_1, \ldots, p_r$ evenly spaced on $[0, n]$
    **for** $l = 1, \ldots, d$ **do**
        Compute $\mathbf{T}^l_{\text{sparse}}\mathbf{x}^l$ with a 1D convolutional filter, size $m$.
        Let $x(t) = \text{sign}(t)\lambda^{|t|}$.
        Form $\mathbf{A}^l \in \mathbb{R}^{r \times r}$ with entries $\mathbf{A}^l_{ij} = k_l(p_i - p_j) = \text{RPE}_l(x(p_i - p_j))$
        Form $\mathbf{W}^l \in \mathbb{R}^{n \times r}$ degree $N$ polynomial interpolation matrix
        Compute $\mathbf{T}^l_{\text{low}}\mathbf{x}^l$ with $\mathbf{T}^l_{\text{low}} = \mathbf{W}^l \mathbf{A}^l \mathbf{W}^{l\top}$
    **end for**
    Return $\text{TNO}(\mathbf{X}) = (\mathbf{T}^1_{\text{sparse}}\mathbf{x}^1 + \mathbf{T}^1_{\text{low}}\mathbf{x}^1, \ldots, \mathbf{T}^d_{\text{sparse}}\mathbf{x}^d + \mathbf{T}^d_{\text{low}}\mathbf{x}^d)$

---

### 3.2.1 SKI For Asymmetric Nyström

Given an asymmetric stationary kernel $k : \mathbb{R} \times \mathbb{R} \to \mathbb{R}$, we wish to approximate the (pseudo) Gram matrix $\mathbf{T} \in \mathbb{R}^{n \times n}$ using a low-rank approximation based on a smaller Gram matrix $\mathbf{A} \in \mathbb{R}^{r \times r}$, with $r \ll n$. In context, $\mathbf{A}$ is formed using relative positions between a set of inducing points $p_1, \ldots, p_r$ instead of the full set $1, \ldots, n$ that is used for $\mathbf{T}$. That is,

$$\mathbf{T}_{ij} = k(i - j) \qquad \text{and} \qquad \mathbf{A}_{ij} = k(p_i - p_j).$$

In our case, the inducing points are uniformly spaced. Some submatrices of $\mathbf{A}$ may be submatrices of $\mathbf{T}$ (if inducing points are also observation points). To derive the Nyström approximation, we form an

augmented Gram matrix $\mathbf{K} \in \mathbb{R}^{(n+r)\times(n+r)}$ in block form as

$$\mathbf{K} = \begin{pmatrix} \mathbf{A} & \mathbf{B} \\ \mathbf{F} & \mathbf{T} \end{pmatrix},$$

where $\mathbf{B} \in \mathbb{R}^{r \times n}$ and $\mathbf{F} \in \mathbb{R}^{n \times r}$ are respectively the upper right and lower left partitions of the large Gram matrix $\mathbf{K}$. Explicitly,

$$\mathbf{B}_{ij} = k(p_i - j) \qquad \text{and} \qquad \mathbf{F}_{ij} = k(i - p_j).$$

Extending [16] to allow singular $\mathbf{A}$,

$$\widehat{\mathbf{K}} = \begin{pmatrix} \mathbf{A} \\ \mathbf{F} \end{pmatrix} \mathbf{A}^\dagger \begin{pmatrix} \mathbf{A} & \mathbf{B} \end{pmatrix} = \begin{pmatrix} \mathbf{A} & \mathbf{A}\mathbf{A}^\dagger\mathbf{B} \\ \mathbf{F}\mathbf{A}^\dagger\mathbf{A} & \mathbf{F}\mathbf{A}^\dagger\mathbf{B} \end{pmatrix}$$

where $\mathbf{A}^\dagger$ is the Moore-Penrose pseudo-inverse satisfying $\mathbf{A}\mathbf{A}^\dagger\mathbf{A} = \mathbf{A}$ (but not necessarily $\mathbf{A}\mathbf{A}^\dagger = \mathbf{I}$ as in [16], which shows up in our different expressions for off-diagonal blocks of $\widehat{\mathbf{K}}$). Following structured kernel interpolation (SKI) [2], we approximate $\mathbf{F}$ and $\mathbf{B}$ using interpolation. Specifically,

$$\mathbf{F} \approx \mathbf{W}\mathbf{A} \qquad \text{and} \qquad \mathbf{B} \approx \mathbf{A}\mathbf{W}^\top$$

where $\mathbf{W} \in \mathbb{R}^{n \times r}$ is a matrix of sparse interpolation weights with up to two non-zero entries per row for linear interpolation or up to four for cubic. These weights can be computed in closed form from the inducing points $p_i$ and the observation points $i$. Thus we have

$$\mathbf{T} \approx \mathbf{F}\mathbf{A}^\dagger\mathbf{B} \approx \mathbf{W}\mathbf{A}\mathbf{A}^\dagger\mathbf{A}\mathbf{W}^\top = \mathbf{W}\mathbf{A}\mathbf{W}^\top$$
$$\Rightarrow \widetilde{\mathbf{T}} = \mathbf{W}\mathbf{A}\mathbf{W}^\top$$

as desired. We can set $\mathbf{T}_{\text{low}} = \widetilde{\mathbf{T}}$ and compute $\widetilde{\mathbf{T}}\mathbf{x}$ by first applying $\mathbf{W}^\top\mathbf{x}$, which is an $O(n)$ operation due to $\mathbf{W} \in \mathbb{R}^{n \times r}$ having sparse rows. Next, we apply $\mathbf{A}(\mathbf{W}^\top\mathbf{x})$. Since $\mathbf{A}$ is a Toeplitz matrix, this is $O(r \log r)$ as per Section 3.1. Finally, $\mathbf{W}(\mathbf{A}\mathbf{W}^\top\mathbf{x})$, the action of $\mathbf{W}$, is again an $O(n)$ operation. Thus computing $\widetilde{\mathbf{T}}\mathbf{x}$ is $O(n + r \log r)$ computation. On a GPU, this factorization achieves a speedup from having small $r$ and being able to leverage efficient parallelized matrix multiplication on specialized hardware. However, in PyTorch [32], we note that for medium sized matrices up to $n = 512$, the time required for data movement in order to perform sparse-dense matrix multiplications can be higher than that of simply performing dense matrix multiplication. This means that in practice, we may instead choose to perform batched dense matrix multiplication, which yields an absolute speedup but a worse asymptotic complexity of $O(nr^2 + r \log r)$.

### 3.2.2 Inverse Time Warp

TNNs use $k_l(i - j) = \lambda^{|i-j|}\text{RPE}_l(i - j)$, where $\text{RPE}_l(i - j)$ is an MLP. There are two issues: 1) the sequential computations required for an MLP are slow, and we only need to evaluate at $2r - 1$ points using SKI instead of $2n - 1$ to produce the full matrix; 2) extrapolation is used in extending to longer sequence lengths than the MLP was trained on, which is generally less reliable than interpolation.

In Proposition 1, we note that an MLP $f : \mathbb{R} \to \mathbb{R}^d$ with ReLU activations and layer normalization is $d$ piecewise linear functions. As we only need to evaluate at $2r - 1$ points, we could let $\text{RPE}_l$ be a piecewise linear function with $r$ grid points. However, we still need to handle extrapolation. We use an inverse time warp and let $\text{RPE}_l$ linearly interpolate on $[-1, 1]$ with the constraint $\text{RPE}_l(0) = 0$ and define $x(t) = \text{sign}(t)\lambda^{|t|}$ for some $0 < \lambda < 1$. We then let $k_l(i - j) = \text{RPE}_l(x(i - j))$.

## 3.3 Frequency Based Approaches

### 3.3.1 Causal Training

The SKI approach allows training bidirectional TNNs with linear complexity. However, fast causal masking negates SKI's benefits (see Appendix B). Thus we need an alternate causal speedup. We use an MLP in the Fourier domain to avoid an explicit time domain decay bias, and use the Hilbert transform to enforce causality. We now describe how we can learn a causal kernel when working in frequency domain (FD). We first define the discrete Hilbert transform, the key tool for achieving this.

**Definition 1.** *The **discrete Hilbert transform** of the discrete Fourier transform $\hat{k}$ is given by*

$$\mathcal{H}\{\hat{k}\} = \hat{k} * h$$

*where $*$ denotes convolution and*

$$h[l] = \begin{cases} 0, & l \text{ even} \\ \frac{2}{\pi l}, & l \text{ odd} \end{cases}$$

The real and imaginary parts of the Fourier transform of a causal function are related to each other through the Hilbert transform. Thus, in order to represent a causal signal, we can model only the real part and compute the corresponding imaginary part. That is, we first estimate an even real function $\hat{k}$ (symmetric about 0) using an MLP. We then take $\hat{k}_{\text{causal}}(\omega) = \hat{k}(\omega) - i\mathcal{H}\{\hat{k}\}(\omega)$.

The inverse Fourier transform $k_{\text{causal}}$ of $\hat{k}_{\text{causal}}$ will thus be causal. For a discussion of why this ensures causality, see [33]. See Algorithm 2 for TNO pseudocode using this approach. Different choices for the smoothness of the frequency domain MLP will lead to different decay rates in time domain, so that smoothness in frequency domain essentially serves the same purpose as the decay bias in [1]. We discuss this theoretically in Section 4.2. Note that we also find that working directly in the frequency domain for bidirectional models (without the Hilbert transform) is often competitive with SKI for speed (despite being $O(n \log n)$ instead of $O(n + r \log r)$) due to needing one fewer FFT.

---

**Algorithm 2** Causal TNO via Discrete Hilbert Transform

---

**Given** sequence $\mathbf{X} \in \mathbb{R}^{n \times d}$ with columns $\mathbf{x}^l$
**Hyperparameters** activation function
**for** $l = 1, \ldots, d$ **do**
    $\hat{\mathbf{x}}^l \leftarrow \mathcal{F}\{\mathbf{x}^l\}$, where $\mathcal{F}$ is the rFFT.
    Compute even real function $\hat{k}^l = \text{RPE}_l(\omega)$, $\omega = \frac{m\pi}{n}$, $m = 0, \ldots, n$.
    Take discrete Hilbert transform $\mathcal{H}\{\hat{k}^l\}$ via the rFFT and irFFT.
    Compute $\hat{k}^l_{\text{causal}}(\omega) = \hat{k}^l(\omega) - i\mathcal{H}\{\hat{k}^l\}(\omega)$ for $\omega = \frac{m\pi}{n}$, $m = 0, \ldots, n$.
    $\mathbf{y}^l \leftarrow \mathcal{F}^{-1}\{\hat{k}^l_{\text{causal}} \odot \hat{\mathbf{x}}^l\}$, where $\mathcal{F}^{-1}$ is the irFFT and $\odot$ denotes an element-wise product.
**end for**
Return $\text{TNO}(\mathbf{X}) = (\mathbf{y}^1, \ldots, \mathbf{y}^d)$

---

### 3.3.2 Bidirectional Training with FD TNN

We extend the FD approach to bidirectional training by removing the causality constraint and model the complex frequency response of real valued time domain kernels directly. To do so we simply double the output width of the RPE and allocate each half for the real and imaginary parts of the kernel frequency responses, while explicitly forcing real-valued responses at $\omega = 0$ and $\pi$. While increasing the complexity of the RPE slightly, we achieve the speed ups in Figure 1 by eliminating the FFTs for the kernels and causality constraint, in addition to the decay bias.

## 4 Theory

We show in Proposition 1 that an MLP mapping from scalars with layer norm and ReLU activations is piecewise linear and continuous, suggesting that using an MLP that we only need to evaluate at a small number of points may be overparametrized, justifying the use of interpolated piecewise linear functions. In section 4.1 we analyze the spectral norm of the matrix approximation error for SKI. We assume the sparse component is exactly identifiable and bound the error of approximating the smooth term via a low-rank SKI factorization. We leave the problem of relaxing this assumption to future work. In section 4.2, we analyze how by using different activations with different smoothness when learning the DTFT of the kernel, we obtain corresponding decay rates for the time domain signal.

**Proposition 1.** *A ReLU MLP $f : \mathbb{R} \to \mathbb{R}^d$ with layer norm and no activation on its output is $d$ piecewise linear continuous functions.*

*Proof.* See Appendix C. □

## 4.1 Matrix Approximation Spectral Norm Error

We give our main error bound for our SKI based low rank approximation. Note that this requires that our kernel is $N + 1$ times continuously differentiable, while the kernel we use in practice uses a piecewise linear function and is thus non-differentiable. In theory, we would need a smoother kernel, adding additional computation overhead. However, we find that empirical performance is still strong and thus we simply use piecewise linear kernels but include the error bound for completeness. Our results depends on the Nyström error $\mathbf{E}_{nyst}$: its $l^2$ norm is bounded in [16].

**Theorem 1.** *Assume that $\mathbf{A}$ is non-singular and $k : [p_1, p_r] \to \mathbb{R}$ is an $N + 1$ times continuously differentiable function, where $p_1$ is the smallest inducing point and $p_r$ is the largest. Let $\mathbf{T}_{r,opt}$ be the optimal rank $r$ approximation to $\mathbf{T}$ and let*

$$\mathbf{E}_{SKI} = \mathbf{WAW}^\top - \mathbf{T}_{r,opt}$$

*be the difference between the SKI approximation using linear interpolation and the optimal one, while*

$$\mathbf{E}_{nyst} = \mathbf{FA}^{-1}\mathbf{B} - \mathbf{T}_{r,opt}$$

*is the difference between the Nyström approximation and the optimal one. Then*

$$\|\mathbf{E}_{SKI}\|_2 \leq \sqrt{nr} \max_{p_{n_1} \leq i \leq p_{n_N}} \frac{|\psi_N(i)|}{(N+1)!} L \left( (N+1)\sqrt{n} + \frac{\min(\sigma_1(\mathbf{F}), \sigma_1(\mathbf{B}))}{\sigma_r(\mathbf{A})} \right) + \|\mathbf{E}_{nyst}\|_2.$$

*where $\psi_N(i) = \prod_{j=1}^N (i - p_{n_j})$ with $p_{n_j}$ being the jth closest inducing point to $i$, $L$ is an upper bound on the $N + 1$th derivative of $k$, and $\sigma_i(\mathbf{M})$ denotes the ith largest singular value of matrix $\mathbf{M}$.*

*Proof.* See Appendix D.1. $\square$

For linear interpolation $\frac{|\psi_N(i)|}{(N+1)!} \leq \frac{h^2}{8}$, where $h$ is the spacing between two neighboring inducing points. We have considered the sparse component of the Toeplitz matrix to be identifiable and focused on the error of approximating the smooth component. While there are potential approaches to relaxing this assumption [34, 35, 36, 37, 38, 39, 40], they must be adapted properly to the Toeplitz setting. Thus, this additional analysis is outside the scope of this paper and a fruitful direction for future work.

## 4.2 Smoothness in Fourier Domain Implies Decay in Time Domain

We now discuss activation function choices when directly learning the discrete time Fourier transform (DTFT) $\hat{k}$ as an MLP. In practice, we sample the DTFT to obtain the actually computable discrete Fourier transform (DFT) by evaluating the MLP with uniform spacing. Different levels of smoothness of the MLP $\hat{k}$ imply different decay rates of the signal $k$. One can think of the choice of activation function as a parametric form for the decay bias. For an MLP, using a GeLU activation implies super-exponential time domain decay. Using SiLU implies super-polynomial time domain decay. For ReLU the signal is square summable. While this subsection focuses on the theoretical relationship between smoothness and decay, in Appendix E.3 we show visualizations demonstrating that these relationships are observed in practice. We first define the DTFT and its inverse.

**Definition 2.** *The **discrete time Fourier transform** [41, 33] $\hat{k}$ or $\mathcal{F}\{k\}$ of $k$ is given by*

$$\hat{k}(\omega) \equiv \sum_{m=-\infty}^{\infty} k[m] \exp(-i\omega m)$$

**Definition 3.** *The **inverse discrete time Fourier transform** of the DTFT $\hat{k}$ is given by*

$$\mathcal{F}^{-1}\{\hat{k}\}[n] \equiv \frac{1}{2\pi} \int_{-\pi}^{\pi} \hat{k}(\omega) \exp(i\omega n) d\omega$$

We now give three theorems relating smoothness of the DTFT to decay of the signal (its inverse).

**Theorem 2.** *Using a GeLU MLP for the DTFT $\hat{k}$, for all $a > 0$, the signal $k[n]$ will have decay*

$$k[n] = O(\exp(-an)).$$

234 *Proof.* See Appendix E.1. □

235 **Theorem 3.** *Using a SiLU MLP for the DTFT $\hat{k}$, the signal $k[n]$ will have decay*

$$|k[n]| \leq \frac{1}{2\pi|n|^N}\big\|\hat{k}^{(N)}\big\|_1$$

236 *for all $n \neq 0, N \in \mathbb{N}$.*

237 *Proof.* See Appendix E.2. □

238 **Theorem 4.** *Using a ReLU MLP for the DTFT $\hat{k}$ implies $\|k\|_2 < \infty$ (the signal is square summable).*

239 *Proof.* Note that $\hat{k} \in L^2[-\pi, \pi]$ since it is continuous. Then apply Parseval's theorem. □

## 5 Experiments

241 We perform experiments in two areas: pre-training a causal language model on Wikitext-103 [8] and
242 training bidirectional models on Long-Range Arena. We start with the repositories of the TNN paper[1]
243 and use their training and hyper-parameter settings unless indicated otherwise. We use A100 and
244 V100s for training, and a single A100 for timing experiments.

### 5.1 Pre-training on Wikitext-103

246 In the causal case we aim to predict the next token, conditional on a fixed length sequence of previous
247 tokens. Table 1 compares FD-TNN's causal pre-training perplexity [8] to existing models: it almost
248 exactly matches that of TNNs. Our approach is faster for the same capacity: at sequence length 512
249 with 6 layer RPEs (as in the TNN paper), FD TNN is 15% faster than the baseline TNN on a single
250 A100 GPU. When both use a three layer RPE, FD TNN is 10% faster. We provide some additional
251 details for this experiment as well as for bidirectional pre-training (we see larger speed gains) in
252 Appendix F.

### 5.2 Long-Range Arena

254 The Long-Range Arena (LRA) is a benchmark with several long sequence datasets. The goal is to
255 achieve both high LRA score (predictive performance) and training steps per second. Following [1],
256 we take the TNN architecture and their tuned hyperparameter (HP) configurations[2], simply replacing
257 their TNO module with our SKI-TNO module with $r = 64$ and $m = 32$. We use $\lambda = 0.99$ where
258 they set $\lambda = 1$, but otherwise perform *no additional HP tuning* on 1D tasks and use smaller layers
259 $r = 32$ and $m = 16$ for the 2D tasks. For FD-TNN, we simply use a same-sized RPE for all tasks
260 except a 3-layer RPE for the CIFAR task. We could potentially achieve even higher accuracy with
261 more comprehensive tuning on the 2D tasks or *any* tuning for the 1D tasks. We select the checkpoint
262 with the highest validation accuracy and report the corresponding test accuracy. SKI-TNN achieves
263 similar average accuracy than TNN at lower size, while FD-TNN achieves *higher* accuracy. We
264 suspect that for some of these problems, the square summable signal implied by ReLU in frequency
265 domain is a better parametric form than applying exponential decay bias. We show our results in
266 Table 2.

267 We additionally perform timing and memory profiling tests on a single 1x A100 instance, keeping
268 the per-GPU batch size constant as in the training runs. In Figure 1a, we plot for each 1D task the
269 percentage of TNN accuracy achieved vs the percentage speedup relative to TNN, with the size of
270 the marker corresponding to the peak memory usage measured. We highlight the 1D tasks because
271 they required no tuning, and they represent the longest sequences at lengths ranging from 1024 to
272 4096, whereas the 2D tasks are treated as separate 1D sequences in each dimension, so that a $32 \times 32$
273 image is seen as alternating length 32 sequences. We note that because the effective sequence lengths
274 are shorter, there is less benefit from using our methods over the baseline TNN.

---

[1]https://github.com/OpenNLPLab/Tnn
[2]https://github.com/OpenNLPLab/lra

| Architecture | PPL (val) | PPL (test) | Params (m) |
|---|---|---|---|
| (Attn-based) | | | |
| Trans | 24.40 | 24.78 | 44.65 |
| LS | 23.56 | 24.05 | 47.89 |
| Flash | 25.92 | 26.70 | 42.17 |
| 1+elu | 27.44 | 28.05 | 44.65 |
| Performer | 62.50 | 63.16 | 44.65 |
| Cosformer | 26.53 | 27.06 | 44.65 |
| (MLP-based) | | | |
| Syn(D) | 31.31 | 32.43 | 46.75 |
| Syn(R) | 33.68 | 34.78 | 44.65 |
| gMLP | 28.08 | 29.13 | 47.83 |
| (SS-based) | | | |
| S4 | 38.34 | 39.66 | 45.69 |
| DSS | 39.39 | 41.07 | 45.73 |
| GSS | 29.61 | 30.74 | 43.84 |
| (TNN-based) | | | |
| TNN (reproduced, 3 layers) | 23.98 (23.96) | 24.67 (24.61) | 48.68 (48.59) |
| FD-TNN: Ours, 3 layers | 23.97 | 24.56 | 48.58 |

Table 1: **Performance on Wikitext-103, Causal Language Model**. We reproduce [1]'s table except for the bottom two rows corresponding to the baseline TNN and our FD-TNN. For both we use the same RPE config with 3 layers. We add in parenthesis the baseline TNN results that we reproduced. We have nearly the same perplexity as the baseline TNN. Our approach is faster: at sequence length 512 with a six layer RPE (as in the TNN paper), FD TNN is 15% faster than the baseline TNN. For a three layer RPE, it is 10% faster.

| Architecture | Text | ListOps | Retrieval | Pathfinder | Image | Avg |
|---|---|---|---|---|---|---|
| TNN | **86.39** | 47.33 | 89.40 | **73.89** | 77.84 | 74.97 |
| SKI-TNN | 83.19 | 45.31 | 88.73 | 68.30 | 76.46 | 72.40 |
| FD-TNN | 85.00 | **55.21** | **90.26** | 69.45 | **84.12** | **76.81** |

Table 2: **Performance on Long Range Arena**. We reproduce experiments and train our proposed variants using tuned hyperparameters from [1]. We **bold** the best and underline the second in each task. Our proposed SKI-TNN and FD-TNN achieve similar overall performance with *no additional hyperparameter tuning* on 1D LRA tasks and a minimal amount of tuning on 2D tasks.

## 6 Conclusion

In this paper, we note that [1]'s Toeplitz neural networks essentially apply the action of a generalized Gram matrix (the Toeplitz matrix) for an asymmetric kernel (the RPE times decay bias) as their main computationally expensive operation. The visualized learned Gram matrices motivate a sparse and low rank decomposition. We thus propose two different approaches to improve efficiency. In the bidirectional setting, we extend SKI to the asymmetric setting and use linear interpolation over a small set of inducing points to avoid the MLP entirely, while using an inverse time warp to handle extrapolation to time points not observed during training. This approach reduces the mathematical complexity from $O(n \log n)$ to $O(n + r \log r)$, where $r$ is the number of inducing points. However in practice, we do not actually use $O(n + r \log r)$ code due to a reshape required for sparse tensors leading to them actually being *slower* than dense tensors. Thus we actually use $O(nr^2 + r \log r)$ in code: still much faster than Baseline TNN for small $r$. For causal training, as causal masking negates SKI's benefits, we instead eliminate the explicit decay bias. We do this by working directly in the frequency domain, enforcing causality via the Hilbert transform and enforcing decay in time domain via smoothness. For the bidirectional case, we eliminate the FFT applied to the kernels. While this maintains $O(n \log n)$ computational complexity, it leads to a substantial speedup in practice and beats TNNs on LRA score.

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
