# Appendix

## Table of Contents

## A  Toeplitz Neural Network Architecture Diagrams

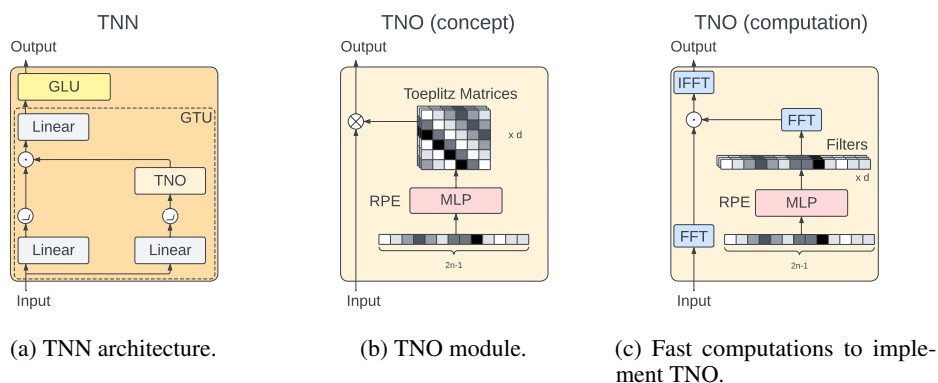

(a) TNN architecture.  (b) TNO module.  (c) Fast computations to implement TNO.

Figure 3: Toeplitz Neural Network and Toeplitz Neural Operators: (a) The overall architecture of a TNN layer [1]. (b) Conceptually, the TNO multiplies each channel of the input by a different Toeplitz matrix. (c) Computationally, the TNO uses FFT's for speed.

## B  Causal Masking negates SKI's benefits

We now show how requiring causal masking for SKI negates its computational benefits on popular hardware accelerators that optimize parallelized matrix multiplication, such as GPUs. Thus, we will need an alternative approach.

First, let's examine the algorithm from [3]. Let $\mathbf{x}' = \mathbf{T}\mathbf{x}$, the subscripted $\mathbf{w}_i \in \mathbb{R}^r$ denote the $i$-th row of $\mathbf{W}$ taken as a column vector, and the subscripted square bracketed $[\mathbf{W}]_i$ denote taking the

434 *i*-th row as a column. That is,

$$\mathbf{x}' = \begin{pmatrix} x_1' & \dots & x_n' \end{pmatrix}^\top \qquad \mathbf{W} = \begin{pmatrix} \mathbf{w}_1 & \dots & \mathbf{w}_n \end{pmatrix}^\top$$

$$\mathbf{x} = \begin{pmatrix} x_1 & \dots & x_n \end{pmatrix}^\top \qquad [\mathbf{W}]_i = \mathbf{w}_i.$$

435 Then

$$x_i' = \sum_{j=1}^i \mathbf{w}_i^\top \mathbf{A} \mathbf{w}_j x_j$$

436 Let us define intermediate sums and resulting recursions,

$$\mathbf{s}_i \triangleq \sum_{j=1}^i \mathbf{w}_j x_j \in \mathbb{R}^r \qquad\qquad \mathbf{s}_i' \triangleq \sum_{j=1}^i \mathbf{A}\mathbf{w}_j x_j \in \mathbb{R}^r$$

$$\Rightarrow \mathbf{s}_{i+1} = \mathbf{s}_i + \mathbf{w}_{i+1} x_{i+1} \qquad\qquad \Rightarrow \mathbf{s}_{i+1}' = \mathbf{s}_i + \mathbf{A}\mathbf{w}_{i+1} x_{i+1}$$

437 so that

$$x_i' = \mathbf{w}_i^\top \mathbf{s}_i' = \mathbf{w}_i^\top \mathbf{A} \mathbf{s}_i = [\mathbf{W}\mathbf{A}]_i^\top \mathbf{s}_i.$$

438 While we *want* to apply the action of $\mathbf{A}$ to $\mathbf{W}^\top \mathbf{x} \in \mathbb{R}^r$ once, which takes $O(r \log r)$. Instead,
439 we *have* to compute one of: (a) $\mathbf{A}\mathbf{s}_i \; \forall i = 1, \dots, n$; (b) $\mathbf{W}\mathbf{A}$; or (c) $\mathbf{A}\mathbf{W}^\top$; all of which take at
440 least $O(nr)$. However, that is not even the largest practical loss. Instead, it is the fact that both
441 cumulative sums $\mathbf{s}_i$ and $\mathbf{s}_i'$ are sequential in nature to compute efficiently (it *is* possible to parallelize
442 the computation with $O(n^2 r)$ memory complexity, also defeating the purpose of this exercise). We
443 found that the sequential nature of the cumulative sum makes it slower than the baseline TNN with
444 FFTs in practice for moderate sequence lengths of at least up to 2048 on current GPUs (NVidia V100,
445 A10, A100). Thus, we need to find an alternate approach for the causal setting.

## C  Proofs Related to Proposition 1

447 We first introduce two auxiliary lemmas, and then prove our main result, which follows immediately
448 from the auxiliary lemmas.

449 **Lemma 1.** *A ReLU MLP $f : \mathbb{R} \to \mathbb{R}$ with no activation on its output is piecewise linear continuous.*

450 *Proof.* Each pre-activation node is a linear combination of piecewise linear continuous functions,
451 and is thus piecewise linear continuous. Each activation applies ReLU, which is piecewise linear and
452 the composition of piecewise linear continuous functions is also piecewise linear continuous. The
453 output is a pre-activation and is thus piecewise linear continuous. □

454 **Lemma 2.** *Adding layer normalization to a ReLU MLP $f : \mathbb{R} \to \mathbb{R}$ preserves piecewise linearity.*

455 *Proof.* Layer normalization applies the same affine transformation to each node in a layer. Since an
456 affine transformation of a piecewise linear continuous function is still piecewise linear continuous,
457 adding layer normalization to an MLP preserves piecewise linear continuity. □

458 **Proposition 1.** *A ReLU MLP $f : \mathbb{R} \to \mathbb{R}^d$ with layer norm and no activation on its output is $d$*
459 *piecewise linear continuous functions.*

460 *Proof.* Follows immediately from Lemmas 1 and 2. □

## D  Proofs for Matrix Approximation Error Spectral Norm

### D.1  Proof of Theorem 1

463 **Theorem 1.** *Assume that $\mathbf{A}$ is non-singular and $k : [p_1, p_r] \to \mathbb{R}$ is an $N + 1$ times continuously*
464 *differentiable function, where $p_1$ is the smallest inducing point and $p_r$ is the largest. Let $\mathbf{T}_{r,opt}$ be*
465 *the optimal rank $r$ approximation to $\mathbf{T}$ and let*

$$\mathbf{E}_{SKI} = \mathbf{W}\mathbf{A}\mathbf{W}^\top - \mathbf{T}_{r,opt}$$

*be the difference between the SKI approximation using linear interpolation and the optimal one, while*

$$\mathbf{E}_{nyst} = \mathbf{F}\mathbf{A}^{-1}\mathbf{B} - \mathbf{T}_{r,opt}$$

*is the difference between the Nyström approximation and the optimal one. Then*

$$\|\mathbf{E}_{SKI}\|_2 \leq \sqrt{nr} \max_{p_{n_1} \leq i \leq p_{n_N}} \frac{|\psi_N(i)|}{(N+1)!} L\left((N+1)\sqrt{n} + \frac{\min(\sigma_1(\mathbf{F}), \sigma_1(\mathbf{B}))}{\sigma_r(\mathbf{A})}\right) + \|\mathbf{E}_{nyst}\|_2.$$

*where $\psi_N(i) = \prod_{j=1}^{N}(i - p_{n_j})$ with $p_{n_j}$ being the jth closest inducing point to i, L is an upper bound on the $N + 1$th derivative of k, and $\sigma_i(\mathbf{M})$ denotes the ith largest singular value of matrix $\mathbf{M}$.*

*Proof.* We first decompose the difference between the SKI approximation and the optimal rank $r$ approximation into the sum of two terms: the difference between the SKI and the Nyström approximations, and the difference between the Nyström and optimal rank $r$ approximations.

$$\begin{aligned}
\mathbf{E}_{SKI} &= \mathbf{W}\mathbf{A}\mathbf{W}^\top - \mathbf{T}_{r,opt} \\
&= \mathbf{W}\mathbf{A}\mathbf{W}^\top - \mathbf{F}\mathbf{A}^{-1}\mathbf{B} + \mathbf{F}\mathbf{A}^{-1}\mathbf{B} - \mathbf{T}_{r,opt} \\
&= \mathbf{W}\mathbf{A}\mathbf{W}^\top - \mathbf{F}\mathbf{A}^{-1}\mathbf{B} + \mathbf{E}_{nyst}
\end{aligned}$$

so that

$$\|\mathbf{E}_{SKI}\|_2 \leq \|\mathbf{W}\mathbf{A}\mathbf{W}^\top - \mathbf{F}\mathbf{A}^{-1}\mathbf{B}\|_2 + \|\mathbf{E}_{nyst}\|_2$$

We need to bound $\|\mathbf{W}\mathbf{A}_M\mathbf{W}^\top - \mathbf{F}\mathbf{A}^{-1}\mathbf{B}\|_2$, the operator norm of the difference between the SKI and the Nyström approximations.

$$\begin{aligned}
\|\mathbf{W}\mathbf{A}&\mathbf{A}^{-1}\mathbf{A}\mathbf{W}^\top - \mathbf{F}\mathbf{A}^{-1}\mathbf{B}\|_2 \\
&= \|\mathbf{W}\mathbf{A}\mathbf{A}^{-1}\mathbf{A}\mathbf{W}^\top - \mathbf{F}\mathbf{A}^{-1}\mathbf{A}\mathbf{W}^\top + \mathbf{F}\mathbf{A}^{-1}\mathbf{A}\mathbf{W}^\top - \mathbf{F}\mathbf{A}^{-1}\mathbf{B}\|_2 \\
&\leq \|\mathbf{W}\mathbf{A} - \mathbf{F}\|_2\|\mathbf{W}^\top\|_2 + \|\mathbf{F}\mathbf{A}^{-1}\|_2\|\mathbf{A}\mathbf{W}^\top - \mathbf{B}\|_2 \\
&\leq \sigma_1(\mathbf{W})\|\mathbf{W}\mathbf{A} - \mathbf{F}\|_2 + \frac{\sigma_1(\mathbf{F})}{\sigma_r(\mathbf{A})}\|\mathbf{A}\mathbf{W}^\top - \mathbf{B}\|_2.
\end{aligned} \tag{1}$$

The first term describes the error due to approximation of $\mathbf{F}$, the left Nyström factor, while the second term describes the error due to approximation of $\mathbf{B}$, the right one. We can use standard interpolation results to bound $\|\mathbf{W}\mathbf{A} - \mathbf{F}\|_2$ and $\|\mathbf{A}\mathbf{W}^\top - \mathbf{B}\|_2$. Recall that the left Nyström factor and inducing Gram matrix have terms

$$\begin{aligned}
\mathbf{F}_{ij} &= k(i, p_j) \\
\mathbf{A}_{ij} &= k(p_i, p_j),
\end{aligned}$$

so that $(\mathbf{W}\mathbf{A})_{ij} = \tilde{k}(i, p_j)$ approximates $\mathbf{F}_{ij} = k(i, p_j)$ using interpolation. For linear interpolation this is

$$\tilde{k}(i, p_j) = w_i k(p_A, p_j) + (1 - w_i)k(p_B, p_j).$$

where $p_A, p_B$ are the two closest inducing points to $i$. More generally with polynomial interpolation of degree $N$ we use $p_{n_1}, \ldots, p_{n_N}$ to denote the $N$ closest inducing points to $i$. Using the Lagrange error formula, polynomial interpolation has the following error bound [42]

$$|\tilde{k}(i, p_j) - k(i, p_j)| \leq \left|\frac{\psi_N(i)}{(N+1)!}\right| \max_{p_{n_1} \leq x \leq p_{n_N}} \left|\frac{\partial^{N+1}}{\partial x^{N+1}} k(x, p_j)\right|$$

where $\psi_N(i) = \prod_{j=1}^{N}(i - p_{n_j})$. As an example, for linear interpolation this gives

$$\begin{aligned}
|\tilde{k}(i, p_j) - k(i, p_j)| &\leq \left|\frac{(i - p_A)(i - P_B)}{2}\right| \max_{p_A \leq x \leq p_B} \left|\frac{\partial^2}{\partial x^2} k(x, p_j)\right| \\
&\leq \frac{h^2}{8} \max_{p_A \leq x \leq p_B} \left|\frac{\partial^2}{\partial x^2} k(x, p_j)\right|,
\end{aligned}$$

where $h = p_B - p_A$ is the distance between any two neighboring inducing points. Note that we assumed the $N + 1$th partial is continuous and since we are interested in $k$ on a compact domain, the $N + 1$th partial is bounded, say by $L$. Thus,

$$|\tilde{k}(i, p_j) - k(i, p_j)| \leq \left| \frac{\psi_N(i)}{(N+1)!} \right| L$$

$$\Rightarrow (\tilde{k}(i, p_j) - k(i, p_j))^2 \leq \left( \frac{\psi_N(i)}{(N+1)!} \right)^2 L^2$$

and thus we can bound the error in the Frobenius norm of the left factor's SKI approximation as

$$\|\mathbf{W}\mathbf{A} - \mathbf{F}\|_F^2 \leq nr \max_{p_{n_1} \leq i \leq p_{n_N}} \left( \frac{\psi_N(i)}{(N+1)!} \right)^2 L^2$$

$$\Rightarrow \|\mathbf{W}\mathbf{A} - \mathbf{F}\|_F \leq \sqrt{nr} \max_{p_{n_1} \leq i \leq p_{n_N}} \frac{|\psi_N(i)|}{(N+1)!} L.$$

This implies an operator norm bound

$$\|\mathbf{W}\mathbf{A} - \mathbf{F}\|_2 \leq \|\mathbf{W}\mathbf{A} - \mathbf{F}\|_F$$

$$\leq \sqrt{nr} \max_{p_{n_1} \leq i \leq p_{n_N}} \frac{|\psi_N(i)|}{(N+1)!} L.$$

The right factor approximation $\|\mathbf{A}\mathbf{W}^\top - \mathbf{B}\|_2$ has the same bound. Plugging into Eqn. 1, we have

$$\|\mathbf{W}\mathbf{A}\mathbf{A}^{-1}\mathbf{A}\mathbf{W}^\top - \mathbf{F}\mathbf{A}^{-1}\mathbf{B}\|_2 \leq \sqrt{nr} \max_{p_{n_1} \leq i \leq p_{n_N}} \frac{|\psi_N(i)|}{(N+1)!} L \left( \sigma_1(\mathbf{W}) + \frac{\sigma_1(\mathbf{F})}{\sigma_s(\mathbf{A})} \right)$$

which gives

$$\|\mathbf{E}_{SKI}\|_2 \leq \sqrt{nr} \max_{p_{n_1} \leq i \leq p_{n_N}} \frac{|\psi_N(i)|}{(N+1)!} L \left( \sigma_1(\mathbf{W}) + \frac{\sigma_1(\mathbf{F})}{\sigma_r(\mathbf{A})} \right) + \|\mathbf{E}_{nyst}\|_2.$$

Now recall that

$$\sigma_1(\mathbf{W}) = \|\mathbf{W}\|_2$$

$$\leq \sqrt{n}\|\mathbf{W}\|_\infty$$

$$\leq (N+1)\sqrt{n}$$

since $\mathbf{W}$ has at most $N + 1$ non-zero entries in each row , so that

$$\|\mathbf{E}_{SKI}\|_2 \leq \sqrt{nr} \max_{p_{n_1} \leq i \leq p_{n_N}} \frac{|\psi_N(i)|}{(N+1)!} L \left( (N+1)\sqrt{n} + \frac{\sigma_1(\mathbf{F})}{\sigma_r(\mathbf{A})} \right) + \|\mathbf{E}_{nyst}\|_2.$$

Note that we could have alternatively expanded Eqn. 1 using terms based on $\mathbf{B}$ instead of $\mathbf{F}$. This gives

$$\|\mathbf{W}\mathbf{A}\mathbf{A}^{-1}\mathbf{A}\mathbf{W}^\top - \mathbf{F}\mathbf{A}^{-1}\mathbf{B}\|_2$$

$$= \|\mathbf{W}\mathbf{A}\mathbf{A}^{-1}\mathbf{A}\mathbf{W}^\top - \mathbf{W}\mathbf{A}\mathbf{A}^{-1}\mathbf{B} + \mathbf{W}\mathbf{A}\mathbf{A}^{-1}\mathbf{B} - \mathbf{F}\mathbf{A}^{-1}\mathbf{B}\|_2$$

$$\leq \|\mathbf{W}\|_2\|\mathbf{A}\mathbf{W}^\top - \mathbf{B}\|_2 + \|\mathbf{W}\mathbf{A} - \mathbf{F}\|_2\|\mathbf{A}^{-1}\mathbf{B}\|_2$$

$$\leq \sigma_1(\mathbf{W})\|\mathbf{A}\mathbf{W}^\top - \mathbf{B}\|_2 + \frac{\sigma_1(\mathbf{B})}{\sigma_r(\mathbf{A})}\|\mathbf{W}\mathbf{A} - \mathbf{F}\|_2.. \tag{2}$$

Using Eqn. 2 instead of Eqn. 1 and taking the min of both results leads to a bound of

$$\|\mathbf{E}_{SKI}\|_2 \leq \sqrt{nr} \max_{p_{n_1} \leq i \leq p_{n_N}} \frac{|\psi_N(i)|}{(N+1)!} L \left( (N+1)\sqrt{n} + \frac{\min(\sigma_1(\mathbf{F}), \sigma_1(\mathbf{B}))}{\sigma_r(\mathbf{A})} \right) + \|\mathbf{E}_{nyst}\|_2.$$

$\square$

## E  Smoothness and Decay

### E.1  GeLU: Proofs Related to Theorem 5

We analyze how modeling the DTFT with a GeLU MLP affects smoothness, the strongest form being an *entire* function, which is complex differentiable everywhere. We then analyze what this implies for the signal. We first recap three basic definitions from complex analysis. In Lemmas 3 and 4, we show GeLU MLPs are entire. In 2 we show that if a DTFT is entire then the signal will decay at faster than any exponential rate. Finally in Theorem 5, we show that modeling the DTFT with a GeLU MLP implies that the signal will decay faster than any exponential rate.

**Definition 4.** *The **complex derivative** of $f : \mathbb{C} \to \mathbb{C}$ at $z_0 \in \mathbb{C}$ is defined as*

$$f'(z_0) = \lim_{z \to z_0} \frac{f(z) - f(z_0)}{z - z_0}.$$

**Definition 5.** *A function $f : \mathbb{C} \to \mathbb{C}$ is **holomorphic** at $z_0 \in \mathbb{C}$ if it is differentiable on a neighborhood of $z_0$.*

**Definition 6.** *A function is **entire** if it is holomorphic on $\mathbb{C}$.*

**Lemma 3.** *The complex extension of the GeLU activation function is entire.*

*Proof.* The GeLU activation function is $x\Phi(x)$, where $\Phi(x)$ is the standard normal CDF. The complex extension is thus $z\Phi(z)$. Recall that

$$\Phi(z) = \frac{1 + \mathrm{Erf}(z/\sqrt{2})}{2}$$

where Erf is the error function. Clearly $z/\sqrt{2}$ is holomorphic on $\mathbb{C}$. It is well known that Erf is holomorphic on $\mathbb{C}$ (see [43] for proof) and compositions of holomorphic functions are holomorphic. Thus $\Phi(z)$ is holomorphic. Finally, the product of holomorphic functions is holomorphic, so that $z\Phi(z)$ is. Since all of this was holomorphic on $\mathbb{C}$, the complex extension of the GeLU activation function is entire. $\qquad\square$

**Lemma 4.** *Each output node of a GeLU MLP with layer norm is an entire function.*

*Proof.* Linear combinations of holomorphic functions are holomorphic, as are compositions. Pre-activations are linear combinations and activations are compositions. The layer-norms are affine transformations, which are also holomorphic. Thus each output node is an entire function. $\qquad\square$

**Proposition 2.** *If the DTFT is entire then*

$$k[n] = O(\exp(-an))$$

*for all $a > 0$.*

*Proof.* Let's consider the Fourier series of $\hat{k}(-\omega)$, which is also entire. Its $n$th coefficient is given by

$$c_n = \frac{1}{2\pi} \int_{-\pi}^{\pi} \hat{k}(-\omega) \exp(-\omega i n) d\omega.$$

Let $u = -\omega$; then $du = -d\omega$ and

$$\begin{aligned} c_n &= -\frac{1}{2\pi} \int_{-\pi}^{\pi} \hat{k}(u) \exp(uin) du \\ &= -k[n]. \end{aligned}$$

Now, Fourier series coefficients for analytic functions in a strip $[-a, a]$ decay as $O(\exp(-an))$. $\quad\square$

**Theorem 5.** *Using a GeLU MLP for the DTFT $\hat{k}$, for all $a > 0$, the signal $k[n]$ will have decay*

$$k[n] = O(\exp(-an)).$$

*Proof.* Follows immediately from Lemma 4 and Proposition 2. $\qquad\square$

 **E.2   SiLU: Proofs Related to Theorem 3**

531   We first argue in Lemma 5 that the SiLU activation function is $C^\infty$. We then show in Proposition 3
532   that SiLU MLPs with layer norm are $C^\infty$ and have integrable derivatives on compact domains. Next
533   in Lemma 6, we argue that for an integrable DTFT, its inverse is bounded by a term proportional to
534   the integral of the DTFT. In Proposition 4, we use the previous lemma to show that the DTFT being
535   $N$ times differentiable implies a decay rate for the original signal. Finally, we prove our main result,
536   that using a SiLU MLP to model a DTFT leads to faster than any polynomial rate in the time domain.

537   **Lemma 5.** *SiLU is $C^\infty$.*

538   *Proof.* The sigmoid function is $C^\infty$, as is the function $x$. The product of $C^\infty$ functions is $C^\infty$.   □

539   **Proposition 3.** *A SiLU MLP mapping scalars to scalars with layer norm is $C^\infty$ with integrable*
540   *derivatives on $[-\pi, \pi]$.*

541   *Proof.* A SiLU MLP with layer norm involves finite linear combinations and finitely many compo-
542   sitions of $C^\infty$ functions, and is thus $C^\infty$. Now any SiLU MLP on a bounded domain has bounded
543   derivatives of all orders (since they are continuous on a bounded domain). Thus, all derivatives are
544   integrable on $[-\pi, \pi]$.   □

545   **Lemma 6.** *If the DTFT $\hat{k} \in L^1[-\pi, \pi]$, then $k$ is bounded and*

$$\|k\|_\infty \leq \frac{1}{2\pi}\|\hat{k}\|_1$$

546   *Proof.* This essentially follows the proof technique of Lemma 9.2.3 in [44], but in the reverse order
547   and using the DTFT instead of the continuous Fourier transform. The idea is to express the signal as
548   the inverse DTFT, which we can since $\hat{k} \in L^1[-\pi, \pi]$, and then use the fact that the values on the
549   complex unit circle have magnitude 1.

$$\begin{aligned}
|k[n]| &= \left| \frac{1}{2\pi} \int_{-\pi}^{\pi} \hat{k}(\omega) \exp(i\omega n) d\omega \right| \\
&\leq \frac{1}{2\pi} \int_{-\pi}^{\pi} |\hat{k}(\omega) \exp(i\omega n)| d\omega \\
&= \frac{1}{2\pi} \int_{-\pi}^{\pi} |\hat{k}(\omega)| d\omega \\
&= \frac{1}{2\pi}\|\hat{k}\|_1
\end{aligned}$$

550   □

551   The next proposition describes how smoothness of the DTFT implies decay of a time domain signal.
552   While there are many very related results in the literature (for instance, [44] shows the opposite
553   direction for the continuous Fourier transform using a very similar proof technique), we were not
554   able to find exactly this result stated or proven rigorously. Thus we state and prove it.

555   **Proposition 4.** *If the Nth derivative of DTFT $\hat{k}$ exists and is integrable on $[-\pi, \pi]$ then*

$$|k[n]| \leq \frac{1}{2\pi|n|^N}\|\hat{k}^{(N)}\|_1$$

556   *for all $n \neq 0$.*

557   *Proof.* We first take the derivative of the DTFT

$$\hat{k}(\omega) = \sum_{m=-\infty}^{\infty} x[m] \exp(-i\omega m)$$

$$\hat{k}'(\omega) = \frac{1}{i} \sum_{m=-\infty}^{\infty} m x[m] \exp(-i\omega m).$$

Since $\hat{k}$ is integrable over $[-\pi, \pi]$, we can plug it into the inverse DTFT

$$\frac{1}{2\pi}\int_{-\pi}^{\pi}\hat{k}'(\omega)\exp(i\omega n) = \frac{1}{2\pi}\int_{-\pi}^{\pi}\frac{1}{i}\sum_{m=-\infty}^{\infty}mk[m]\exp(-i\omega m)\exp(i\omega n)d\omega$$

$$= \frac{1}{i}\sum_{m=-\infty}^{\infty}mk[m]\delta[n-m]$$

$$= \frac{n}{i}k[n]$$

so that if $\hat{k}$ and $\hat{k}'$ are integrable, we obtain the key identity relating the inverse DTFTs of a DTFT and its derivative

$$\mathcal{F}^{-1}\{\hat{k}\} = \frac{i}{n}\mathcal{F}^{-1}\{\hat{k}'\}. \tag{3}$$

Thus

$$|k[n]| \leq \frac{1}{|n|}\left|\mathcal{F}^{-1}\{\hat{k}'\}[n]\right|$$

$$\leq \frac{1}{n^2}\left|\mathcal{F}^{-1}\{\hat{k}^{(2)}\}[n]\right| \qquad \text{Eqn. 3, since } \hat{k}^{(2)} \text{ integrable}$$

$$\leq \frac{1}{|n|^N}\left|\mathcal{F}^{-1}\{\hat{k}^{(N)}\}[n]\right| \qquad \text{applying recursively, since } N\text{th derivative integrable}$$

$$\leq \frac{1}{2\pi|n|^N}\left\|\hat{k}^{(N)}\right\|_1$$

where the last line follows from Lemma 6. $\qquad\square$

**Theorem 3.** *Using a SiLU MLP for the DTFT $\hat{k}$, the signal $k[n]$ will have decay*

$$|k[n]| \leq \frac{1}{2\pi|n|^N}\left\|\hat{k}^{(N)}\right\|_1$$

*for all $n \neq 0, N \in \mathbb{N}$.*

*Proof.* This follows immediately from Proposition 3 and Proposition 4. $\qquad\square$

### E.3 Visualizations for Smoothness and Decay

We visualize the frequency responses and the corresponding impulse responses generated by the frequency domain (FD) RPE under the three activation functions for which we have shown theory, with results predicted by theory. For a randomly initialized FD RPE with Gelu activations the impulse responses decay to approximately 0 by $n = 30$: this is very rapid decay and the curves visually look like exponential decay. For a randomly initialized SiLU RPE, the resulting impulse responses are similar. For the ReLU case we show the generated filters from a trained FD TNN RPE from one of the TNN layers. We see the impulse responses visually decay to approximately 0 within the finite length of 512 points. This is a slower rate of decay than either of the previous two.

## F   Experiment Details and Additional Results

### F.1   Wikitext-103

#### F.1.1   Fourier Domain

For both causal and bidirectional models we use the default model and training hyperparameters from the TNN repository as the TNN baseline, defined in the first two columns in [1] Table 13: LM (causal) and Roberta (bidirectional). One small HP discrepancy between the repository and table is the use of 7 decoder layers for the causal LM, which we used for all LM experiments, instead of the 6 they had in their paper. We find that we can reduce the default number of RPE layers from 6 to 3 and improve the speed of the baseline with slight quality improvements. We provide these reproduced

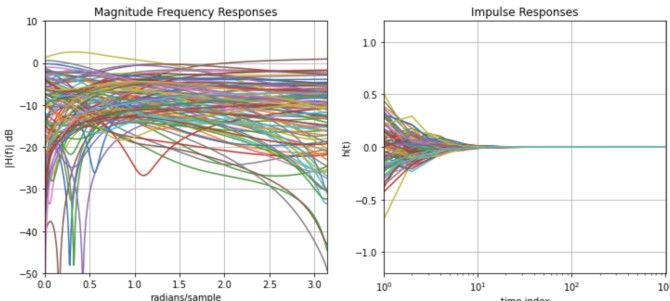

Figure 4: Frequency and impulse responses for a randomly initialized FD RPE MLP with **GeLU** activations. The curves on the left side are holomorphic, and theory predicts that the curves on the right hand will decay at faster than any exponential rate. They appear to decay approximately exponentially.

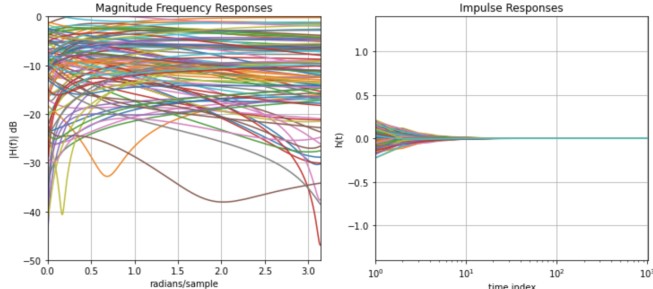

Figure 5: Frequency and impulse response for a randomly initialized FD RPE MLP with **SiLU** activations. The curves on the left side are $C^\infty$, and theory predicts that the curves on the right will decay at faster than any polynomial rate. They appear visually to have 'almost' exponential decay.

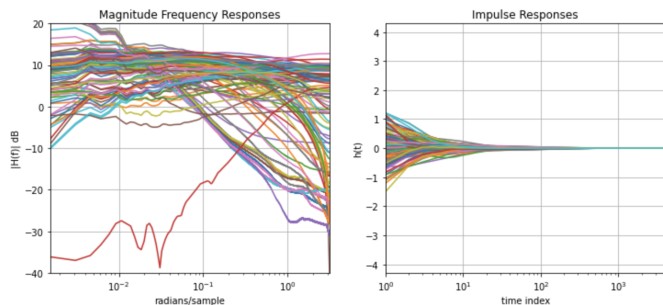

Figure 6: Frequency and impulse responses from an FD RPE MLP with **ReLU** activations, taken from one layer of a trained FD TNN. The curves on the left are continuous, and theory predicts that the curves on the right will be square summable. They clearly will vanish at infinity, although it is not immediately visually clear at what rate.

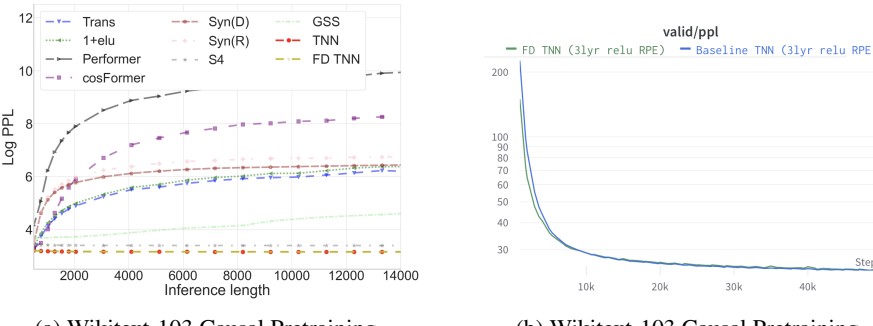

(a) Wikitext-103 Causal Pretraining.

(b) Wikitext-103 Causal Pretraining.

Figure 7: a) In Wikitext-103 causal pretraining, our approach, FD TNN achieves equivalent perplexity vs inference length to TNN. b) Validation Perplexity vs iterations. In the causal setting, FD TNN converges to an equivalent quality at the same rate, but with a 5 to 15% increase in training speed depending on the RPE MLP depth (see Figure 1). For these experiments, we used a learning rate 1e-3 for FD TNN and the default (5e-4) for the baseline.

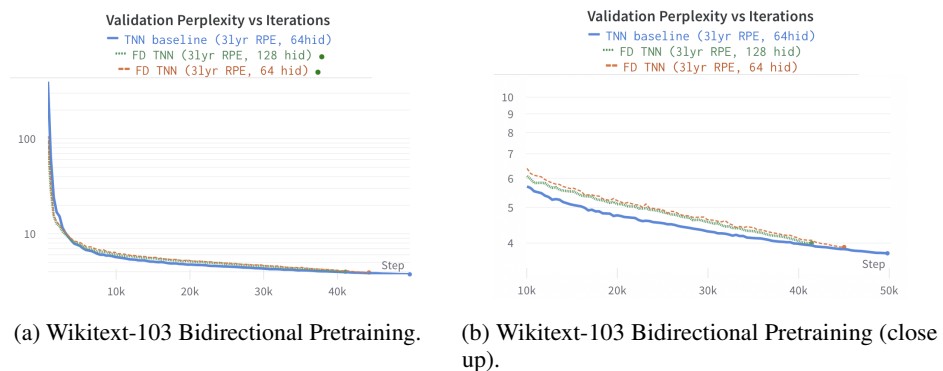

(a) Wikitext-103 Bidirectional Pretraining.

(b) Wikitext-103 Bidirectional Pretraining (close up).

Figure 8: a) In Wikitext-103 bidirectional pretraining, after minimal HP tuning from the default, we observed that FD TNN slightly lags the validation perplexity of the TNN baseline throughout much of the 50k training iterations, but closes this gap during the last 10k iterations. As a result, our 35-80% speed up in iterations/sec (Figure 1b) applies to wall clock time assuming one trains for approximately 50k steps. For these results, we used a learning rate of 1e-3 for FD TNN and the default (5e-4) for the baseline.

perplexity scores for the baseline in parenthesis in Table 1, next to those reported by [1]. For causal pretraining at a 512 sequence length, FD TNN achieves equivalent perplexity vs inference length as the TNN baseline (see Figure 7a). We achieve between a 5 and 15 % speed up for the causal case, and a nearly 80 % speed up in the best case (6 RPE layers) for the bidirectional case.