# OpenReview forum: "SKI to go Faster: Accelerating Toeplitz Neural Networks via Asymmetric Kernels"
_NeurIPS.cc/2023/Conference — Submitted to NeurIPS 2023_

### Official Review · Reviewer_fqtA · 2023-06-16

**Soundness:** 2 fair
**Presentation:** 2 fair
**Contribution:** 2 fair
**Rating:** 5
**Confidence:** 3

**Summary:**

The present paper offers a Toeplitz matrix architecture which can handle sequence modeling. The architecture comes in two flavors. The first flavor is a fast version that is most useful for bi-directional tasks. It speeds up previous Toeplitz networks by using an interpolation and low rank approximation scheme in its setup. The second is a Fourier based model that appears to offer advantages in causal tasks. Benefits are shown in terms of the speed of training and some marginal benefits in performance.


**Strengths:**

The main strength of this paper is a noticeable speed up in an alternative architecture to transformers. I am a fan of papers that look at ways to speed up sequence modeling. The Curren paper presents a nice idea. More specifically:
- The approximations to the TNN appear to be effective in experiments and result in a faster network
- Approximations do not seem to deteriorate performance and may offer some added performance boosts
- The changes to the architecture are grounded in some theory


**Weaknesses:**


**Disclaimer**: I am not an NLP expert and am more focused on the theory side. I have significant concerns about the theory in this paper, but feel the experiments and techniques are solid enough to potentially overcome that issue. Furthermore, none of the theorems are crucial to the crux of the paper, and if any are wrong, they can be removed. For this reason, I placed a borderline accept rating for now, but I believe this will need to be confirmed by people who are closer to the experimental side of the literature and can assess the experiments in a more rigorous fashion.

\
Broader comments:
- Most of my larger concerns revolve around the theory and proofs in this project listed below.
- Reading through many times, I could not understand what was gained in the “causal training” proposal in section 3.1. First, it seems the parameters are changed to live completely in the Fourier regime. This change was made to obtain “an alternate causal speedup", but I don’t see where that speedup arises. Following the steps, the main change seems to take the algorithm to do an FFT on the $n$-dimensional space resulting in runtime of $O(n \log n)$ which is worse than before. Also, changing the algorithm to work in Fourier space introduces a different implicit bias that I’m not sure is desired. For example, the $\lambda$ parameter controlling the decay is not controlled here. I also have many questions about this approach which I’ve left below.
- Experiments in table 1 don’t appear to offer much improvement especially considering the added parameters. I would also ask the authors to include citations to the models or results compared to in this table so it is easier to see what is being compared to.

\
Theory comments:
- Theorem about ReLU MLPs being $d$-piecewise linear has assumptions missing or is just wrong. If the authors are implying that any ReLU network from $\mathbb{R} \to \mathbb{R}^d$ has $d$ pieces or contiguous linear regions, this is clearly wrong. MLPs are universal approximators so this is clearly false. If the authors are saying that this only holds for an MLP with a single hidden layer of width at most $d$ then this may be correct. But I don’t see this assumption made anywhere.
- Theorem 2 has a few confusing elements from my end. First, it is hard to parse. There are many variables and factors like condition number that it is hard to know the scaling of. Second, the bound doesn’t appear to be all that good. The error grows at least linear in $n$ and depends on other factors like singular values or the nystrom error that may also be badly bounded. I suppose the authors would argue that it is exponentially small in the degree of interpolation $N$, but this degree would have to grow at least logarithmically in $n$ to counteract the $n$ factor. This would result in a runtime essentially equivalent to just doing FFTs on the whole space. Third, the error in interpolation is an unusual thing to even bound in my opinion. The weights are updated with this interpolation taken into account. In other words, the algorithm learns a weight matrix with parameters contained in this interpolation.
- Definition 2 and 3 present the discrete time Fourier transform, but in practice only the DFT of the matrix form is ever used. The resulting statements, regardless of their correctness, do not seem to apply to the setting in practice. Unless I am missing something.
- Related to the above, I cannot see why Theorem 3 and 4 are correct. Similar to my previous statements, MLPs are universal approximations so they can output any possible function. How can that statement hold true?

\
Small:
- Simply having a % label on the y axis of Fig 1B is confusing. Percentage relative to what? Also, what does 20% speed-up mean; i.e. that it ran in 20% less time? Simply having this number on the first page can be confusing without the context added.
- Fig 1A and 1B also appear to be different size fonts.
- Line 150: I think the dense-case runtime is only a factor of $r$ worse so $O(nr  + r \log r)$ and not $O(nr^2 + r \log r)$ unless I’m missing something.
- For someone outside of the NLP community, section 3.1 needed more motivation and formality. Some details about what “causal masking”, “causal kernel”, and the sequential nature of the data would be helpful.
- To follow easier, it would be good to define what the role of N is in section 4.1 (i.e., number of interpolating points)

\
Formatting:
- Hyperref links seem to be broken
- Line 189: sentence is a run-on and hard to follow

\
Finally, to add ideas not for criticism, but instead for completing the paper and offering new ideas, there is a wealth of literature on related techniques that could be useful here, or at the very least cited. For example, there are sparse Fourier transforms that can offer speed-ups beyond the $O(n \log n)$ that the paper aims to improve on, e.g., [1]. Since matrices are low-rank and/or sparse, this could be a more direct way to get the speed-ups desired. Second, there are a number of papers on optimizing structured matrices like Toeplitz matrices, e.g. [2]. In the sequence modeling specifically, there have been a lot of papers on unitary networks for example [3], one paper which actually uses low rank approximations in its implementation [4]. Third, low rank approximations have also been used to speed up other architectures like conv-nets, [5-6]


\
**References:** \
[1] Hassanieh, Haitham, et al. "Simple and practical algorithm for sparse Fourier transform." Proceedings of the twenty-third annual ACM-SIAM symposium on Discrete Algorithms. Society for Industrial and Applied Mathematics, 2012.\
[2] Kochurov, Max, Rasul Karimov, and Serge Kozlukov. "Geoopt: Riemannian optimization in pytorch." arXiv preprint arXiv:2005.02819 (2020).\
[3] Arjovsky, Martin, Amar Shah, and Yoshua Bengio. "Unitary evolution recurrent neural networks." International conference on machine learning. PMLR, 2016.\
[4] Kiani, Bobak, et al. "projUNN: efficient method for training deep networks with unitary matrices." arXiv preprint arXiv:2203.05483 (2022).\
[5] Max Jaderberg, Andrea Vedaldi, and Andrew Zisserman. Speeding up convolutional neural networks with low rank expansions. arXiv preprint arXiv:1405.3866, 2014.\
[6] Cheng Tai, Tong Xiao, Yi Zhang, Xiaogang Wang, et al. Convolutional neural networks with low-rank regularization. arXiv preprint arXiv:1511.06067, 2015.



**Questions:**

Beyond what's written above, I have the following questions:
- How do you ensure the complex valued MLP gives real valued outputs in algorithm 2 after the iFFT?


Questions about causal training:
- Why do we need a Hilbert transform when we can just do a regular FFT on a standard Toeplitz matrix and it results in the same runtime?
- Is the $\lambda$ factor which decays farther out correlations just ignored here?
- Aren’t there much easier ways to enforce a causal filter? E.g., only have the Toeplitz matrix values nonzero along the left of the diagonal of the matrix.

**Limitations:**

There is a very brief discussion of limitations in the conclusion, though I feel this could be expanded. I would also appreciate some context for this work in relation to other works in NLP and how it fits into the broader landscape of NLP architectures. For someone like me not in the community, this would be useful to better understand its limitations from a practical perspective.

---

> ### Author Rebuttal · Authors · 2023-08-03
>
> Thank you for your helpful critiques and suggestions. We will use them to improve clarity.
>
> # Weaknesses
> ## Broader Comments
> * Speedups arise from not multiplying the RPE by a decay bias.
> * Table 1: FD-TNN doesn't add parameters (48.58 million vs TNN's 48.59). We are faster than TNNs with similar performance. TNN/FD-TNN perplexity is *much* lower than all approaches other than transformers/LS, which are slower (Figure 1 of [3]).
> * We will add model citations in an appendix.
>
> ## Theory Comments
>
> ### On piecewise linear functions
> Our statement doesn't claim $d$ linear regions: indeed this would be false. Rather we have $d$ separate piecewise linear functions. That is, $f:\mathbb{R}\rightarrow \mathbb{R}^d$ is a collection of functions $f_i:\mathbb{R}\rightarrow \mathbb{R},i=1,\ldots,d$. Each $f_i$ is a piecewise linear function that may have many pieces. We will emphasize this distinction in the paper. Piecewise linear functions from $\mathbb{R}$ to $\mathbb{R}$ are also universal approximators of continuous functions on intervals [1]: we have $d$ of them.
>
> ### On Theorem 2
>
> Note the error beyond Nyström is *not* exponentially small in $N$: it is factorially small (under well behaved $\psi_N,L$). Rather than needing $N=O(\log n)$, instead $N\log N+(N-1)=O(\log n)$ suffices: a much weaker condition. To see why, note $N!\geq \frac{N^N}{\exp(N-1)}$ and apply the same technique used to derive that $N=O(\log n)$ if the error decayed exponentially with $N$.
>
> The bound *is* somewhat loose due to analyzing how much worse than Nyström it is, rather than analyzing SKI's interpolation error directly. The dependence on $\frac{1}{\sigma(A)}$ comes from approximating Nyström with the interpolated left factor, inherited from the asymmetric Nyström analysis of [2].
>
> Our theory yields different intuition compared to that of other kernel methods due to our asymmetric setting. For SPD kernels, results are typically expressed in terms of the kernel function's properties (eigenvalues etc). In the asymmetric setting, matrix norms and singular values of matrices provide a discretized/implicit view of the kernel function's properties. Despite these limitations, this is the _first_ theoretical result for SKI, and it holds for the asymmetric setting as well. We will clarify these intuitions in the text. In future work we plan to derive a stronger result.
>
> ### On DTFT vs DFT
>
> In signal processing, the DTFT and DFT are known to be intimately related [4]. Consider a continuous frequency domain signal $X(w)$ with domain $w$ in $[0, 2\pi]$. Taking the inverse DTFT yields the infinite-length discrete time signal $x[n]$. In practice we evaluate $N$ discrete frequencies $X'[k]=X(2\pi k/ N)$ and take inverse DFT to produce $x'[n]$. Then $x'[n]$ is an _aliased_ version of $x[n]$. The decay rate of $x[n]$ can tell us how good of an approximation $x'[n]$ actually is to $x[n]$. As the rate of decay increases, $x'[n]$ becomes a better estimate of $x[n]$; or, for a fixed decay rate, as $N \rightarrow \infty$, the effect of aliasing decreases, so $x'[n]$ becomes a better estimate of $x[n]$. We will expand our discussion to include this clarification.
>
> ### On Theorems 3 and 4
>
> Indeed MLPs are universal approximators. However we don't state that the MLP must decay; rather if the MLP is smooth, then its inverse Fourier transform will have a corresponding decay rate. Note smoothness does *not* preclude universal approximation. The connection between smooth functions in one domain (time or frequency) and decay in the other is well studied in Fourier analysis, but the most well-known results are for the continuous Fourier transform on $\Omega \in (-\infty, \infty)$ instead of the DTFT on $\omega \in (-\pi, \pi]$.
>
> ## Small
> * Yes, it ran in 20\% less time. We will clarify this.
> * We will fix this.
> * The $O(nr^2)$ comes from $WA$ being an $n\times r$ matrix times an $r\times r$ one. Thus we have $r$ matrix vector multiplications: each cost $O(nr)$, total complexity is $O(nr^2)$.
> * We will add this. Causal masking refers to what you described: taking the lower triangular matrix, i.e. in a causal kernel if $j>i$, then $k(i,j)=0$.
>
> ## Your Suggestions
>
> These are helpful, thanks. We looked at sparse FFTs. As the frequency representation is not truly sparse, we found that sparse frequency representations hurt performance. Further, the more specialized methods would likely require sophisticated custom implementations to actually run faster than Pytorch's more generic functions. For example, we also tried using (Kochurov 2020) initially, but it was slow (perhaps because of its power/generality). It is worth exploring how to incorporate unitary matrices into this approach. We will cite several of these papers and think about them as future extensions.
>
> # Questions
>
> * We enforce a real valued time kernel by forcing real-valued DC and Nyquist frequencies.
> * This approach avoids $\lambda$ as you mention, lending speedups due to not multiplying the RPE by a decay bias.
> * One no longer needs $\lambda$: the decay rate is implied by the MLP's smoothness, with learned constants.
> * Yes, but using an MLP directly in frequency domain allows us to let the MLP's smoothness imply a decay rate with learned constants.
>
> # Limitations
>
> Thank you. We will add discussion relating this to long convs and long convs to transformer/xformers, including a summary of respective empirical strengths on benchmarks and modeling tradeoffs.
>
> [1] We can't link websites; a search for 'Piecewise linear function close to continuous function' yields the proof.
> [2] Nemtsov, Arik, et al. "Matrix compression using the Nyström method." Intelligent Data Analysis 2016.
> [3] Qin, Zhen, et al. "Toeplitz Neural Network for Sequence Modeling." 11th International Conference on Learning Representations 2022.
> [4] Proakis, JG and Manolakis, DG. "Introduction to Digital Signal Processing." Prentice Hall Professional Technical Reference, 1988.

---

> > ### Comment · Reviewer_fqtA · 2023-08-10
> > **Continuing discussion on theory**
> >
> > Dear authors,
> >
> > Thank you for your response. I want to stress that I feel your experiments and basic idea is nice, but I still have serious concerns about the theory. In fact, as it stands now, I would support removing formal statements and replacing themwith heuristic justifications since I still do not believe the statements that you claim are rigorously proven.
> >
> > Let's take these in order:
> > - **Proposition 1:** Thank you for clarifying your meaning of this proposition. However, this is a rather trivial statement that I don't think needs a proof. It is obvious that ReLUs form piecewise linear mappings and this does not need a proof. Furthermore, $d$ piecewise linear mappings is just the same as a single piecewise linear mapping so what is that statement even telling us?
> > - **Theorem 2:** Thank you for your response. Factorial growth and exponential growth are the same up to logarithmic factors in the exponent, so this does not change the story in my opinion. This would just reduce the $N$ dependence by a factor $O(\log ( \log (n)))$. More broadly, I am still not sure whether or how all of the other parameters in the theorem are bounded. There is a dependence on a number of factors and even the asymptotics are hard to garner. It still appears that $N$ needs to be taken as something like $\tilde O(\log n )$ (hiding sub-logarithmic contributions). Thus, I am not convinced that my original point is incorrect that this is not much better than just using an FFT.
> > - **Theorems 3 and 4:** I am confused by your answer. Yes, I agree it is possible to approximate a discrete FFT with samples from a continuous one, but here, your inputs and outputs are necessarily discrete. In all settings, you are literally using the discrete FFT. So a statement like taking $N \to \infty$ or "consider a continuous frequency domain signal" makes no sense to me. A rigorous proof would consider exactly the setting you are looking at, not some continuous setting.
> >
> > As it stands, too many changes are needed and proofs cannot be checked again with the current format of these reviews. I am afraid I cannot accept the statements as rigorous mathematical proofs. Hence, I feel the authors should remove these mathematical statements, replace them with heuristic justifications, and remove confusing assumptions and claims.
> >
> > Again, I support the idea in this paper and like the experiments, but the theory is just not rigorous enough for me.

---

> > > ### Author Response · Authors · 2023-08-12
> > > **Response**
> > >
> > > Thank you for your thoughts. We believe that there is a difference between rigorous and tight. We do provide rigorous proofs for our theoretical statements and interpret them in the context of our method. To the extent that there is a gap between theory and practice, or that our bounds could be stronger, we do not claim that our results are tight, and we mention that future work remains in improving the theoretical statements. We review the specific areas with theoretical critiques:
> > >
> > > **Proposition 1**: We provided this statement for completeness. Although some may consider the result trivial, it is a short statement to help justify use of linear interpolation. We can however move the formal statement to an appendix, and only briefly mention it in the main paper.
> > > **Theorem 2**: While this is admittedly not a tight result, it is the first. It justifies our use of the approximation: having a bound lets us know that our error will not be arbitrarily bad. In practice, our empirical results show that we are able use a lower value for $N$ than would result in computation equivalent to the FFT and thus achieve comparable performance with a sizeable speedup.
> > > **Theorems 3 and 4**: We thank the reviewer again for highlighting the nuances of this point. While it is true we can only compute on discrete samples, it is nonetheless meaningful to consider the continuous frequency domain signal because the underlying function being represented by the network is continuous. In addition, since we consider extending the sequence length post-training, we are in fact exactly concerned with taking $n\rightarrow\infty$. Note that this gap between finite sample in practice and large sample in theory is not a limitation unique to our setting, but is also core to asymptotic statistics, the study of infinite width neural networks and many other areas. While we often prefer finite sample guarantees to asymptotic ones, asymptotic ones are an important first step towards understanding a problem.
> > >
> > > Rather than changing the proofs, as we believe that they are correct as is, we propose adding a small bit of clarifying text in the main body to acknowledge any gaps between theory and practice, as we have suggested in these discussions.

---

> > > > ### Comment · Reviewer_fqtA · 2023-08-14
> > > >
> > > > Dear authors,
> > > >
> > > > Let's set aside proposition 1 and Theorem 1 for now. At the very least, with those two statements, significant clarification is needed in the text to explain why they are there and how tight the bounds are. I can have faith that you will make those changes.
> > > >
> > > > I still do not follow any of your reasoning for Theorems 2 to 4. Let me ask some clarifying questions.
> > > >
> > > > The $k \in \mathbb{R}^{2n+1}$ that you are studying corresponds to the function discretized at $2n+1$ points which provides values for the Toeplitz matrix $T_{ij} = k(i-j)$. Let us treat $k$ as a vector to make things easier. To speed up Toeplitz matrix calculations, an FFT is performed at some point to a version of the matrix which is made to be circulant.  Here, this results in something like
> > > > $$FFT \cdot k [a] = \hat{k}[a] = \sum_{b=0}^{2n} k(b-n) \exp(-2 \pi iab/(2n+1)),$$
> > > > where in the above we treat $\hat{k}$ also as a vector.
> > > >
> > > > The discrete time Fourier transform, in contrast, takes in a sequence $k':\mathbb{Z} \to \mathbb{R}$ (i.e., an infinite length sequence). Here, the resulting discrete time Fourier transform is
> > > > $$DTFT \cdot k' [a] = \hat{k}'[a] = \sum_{b=-\infty}^{\infty} k'(b) \exp(-2 \pi iab).$$
> > > > Note, that conventions may differ form the above, but the key point is that the above takes in a function valued on all integers. The DTFT can be sampled into a discrete form, but this requires taking  a periodic summation of the signal, something like $k_{N}[n] = \sum_{m=-\infty}^{\infty} k'[n-mN]$, which you are explicitly not doing.
> > > >
> > > > So as far as I can tell, if you want to equate the discrete time Fourier transform with the DFT, there is a tail in the sequence that must be bounded. I.e., the discrete time Fourier transform is also supported in the range $(\infty, -n) \cup (n, \infty)$ which causes an error with respect to the DFT.
> > > >
> > > > As a second point regarding asymptotics. It is ok to use asymptotic analysis, but the order in which you send things to infinity matters. Here, you need to fix a kernel (i.e. whatever is learned), analyze the error of that kernel with respect to the discrete time Fourier transform, send the input dimension to infinity, and then the frequency to infinity. This does not appear to be an easy thing to do. Sending the frequency to infinity for any fixed input dimension makes no sense based on the above discussion, since the DFT (which is what matters in practice), is only supported on a finite window size.

---

> > > > > ### Author Response · Authors · 2023-08-14
> > > > > **Response**
> > > > >
> > > > > Thanks for the question. We will clarify our prior argument with notation to (mostly) match yours now that space allows: we use square brackets to denote discrete time arguments, parentheses to denote continuous time arguments, and curly braces to denote the argument of transforms. As we alluded to in our initial response and in our paper, it is more straightforward to consider the _inverse_ direction of the transform. We are still working with a fixed learned kernel as you have correctly recognized as necessary; however, we start with the fixed kernel _in the frequency domain_  rather than the time domain as your analysis attempts. This avoids the difficulties you ran into and pointed out in your last paragraph.
> > > > >
> > > > > We represent an ideal fixed learned kernel $k[n]$ via its continuous frequency domain signal $\hat{k}(w)$ with domain $w$ in $[0, 2\pi)$, which is in practice what we actually represent with the neural network. Thus, we start with $\hat{k}(w)$ instead of $k[n]$. Taking the _inverse_ DTFT yields the infinite-length discrete time signal
> > > > > \begin{align*}
> > > > > k[n] = \textrm{IDTFT} \\{\hat{k}(w)\\},
> > > > > \end{align*}
> > > > > which we never get to observe. Instead, in practice we evaluate our neural network $\hat{k}(w)$ at $N$ equally spaced discrete frequencies in the interval $w \in [0, 2\pi)$ with $m=0,\ldots,N-1$ (decreasing the space between samples as $N \rightarrow\infty$), i.e.
> > > > > \begin{align*}
> > > > > \hat{k}'[m]=X(2\pi m/ N),
> > > > > \end{align*}
> > > > > which is the FFT of some approximate kernel $k’[n]$, i.e. taking _inverse_ DFT produces
> > > > > \begin{align*}
> > > > > k'[n] = IDFT\\{\hat{k}’[m]\\}.
> > > > > \end{align*}
> > > > > Then $k'[n]$ is indeed an aliased version of $k[n]$ as you also allude to. This means that $k’[n]$ can be seen as a periodic infinite length sequence that we only need the $N$ points $n=0,\ldots,N-1$ to represent. As a result, we do not need an explicit “periodic summation of the signal” because the representation does so implicitly.
> > > > >
> > > > > The decay rate of $k[n]$ can tell us how good of an approximation $k'[n]$ actually is to $k[n]$ (on the initial $N$ points; we will omit this in the remainder). As the rate of decay increases, $k'[n]$ becomes a better estimate of $k[n]$; or, for a fixed decay rate, as $N \rightarrow \infty$, the effect of aliasing decreases (a dirac comb with decreasing spacing produces an inverse Fourier transform with increasing spacing), so $k'[n]$ becomes a better estimate of $k[n]$.
> > > > >
> > > > > This corresponds to fixing a frequency domain representation, taking the discretization frequency to infinity, and computing the corresponding _inverse_ DFT for a given fixed input dimension.

---

> > > > > > ### Comment · Reviewer_fqtA · 2023-08-15
> > > > > >
> > > > > > My apologies for misunderstanding the representation of the function. Now, I understand that the function learned is in the Fourier domain. Just to confirm, the MLP is used a function $f: [0,2\pi] \to \mathbb{R}$, then sampled at points $[0, \pi/n, \dots, 2 \pi]$, and then the $n$ points are treated as a vector in $\mathbb{R}^n$ to which an inverse FFT is applied?
> > > > > >
> > > > > > If that is correct, the same issue applies as far as I can tell. The $IDFT$ function that you write only inverts elements in the finite support indexed by the $n$ points. So sending $n \to \infty$ and fixing the MLP still does not make sense to me.
> > > > > >
> > > > > > For this analysis to be precise, the order of operations must be taken in a proper order. You want to know the value of the MLP in its tail, let's say at a point $x(n)$ which grows with $n$ (e.g., the last indexed point in the sequence). I.e. we want to know the growth or decay of $k(x(n))$ for the learned MLP. Here is how it should be done from my understanding:
> > > > > > - For every $n$, you are studying a distinct problem with a distinct dataset and MLP. Therefore, for each $n$, you learn a $\hat{k}_n: [0,2\pi] \to \mathbb{R}$ which corresponds to the kernel in frequency space. To make this clear, let us denote this as a sequence $\hat{k}_n$. Here, $n$ denotes the sequence length, i.e. the same $n$ in your algorithm 1 and 2.
> > > > > > - The, you study the limit $\lim_{n \to \infty} k_n(x(n)) = \lim_{n \to \infty} IDFT( \hat{k_n})  (x(n))$. Or alternatively, you take the limit $\hat{k}=\lim_{n \to\infty} \hat{k_n}$ to find the limiting function (if it exists) and then proceed to do asymptotic analysis of this.
> > > > > >
> > > > > > During the above procedure, you need to account for errors in the aliasing as you call it since the value of the function at $x(n)$ from the inverse FFT is not the same as that given by the inverse discrete time Fourier transform.
> > > > > >
> > > > > > In contrast, in all of your proofs in Theorem 2 to 4, you are fixing a single MLP for all $n$ and then studying the limit which does not make sense to me.
> > > > > >
> > > > > > Again, I want to stress, that I like your approach overall, and believe your mathematical theory can serve as heuristic justification. However, I remain unconvinced by the rigor of the proofs and still cannot take them as formal statements.

---

> > > > > > > ### Author Response · Authors · 2023-08-15
> > > > > > >
> > > > > > > Thanks for clarifying. We believe that you are still describing a _slightly_ different setting from what we do. Specifically, you said that “For every $n$, you are studying a distinct problem with a distinct dataset and MLP.” Recall that the setting of interest is for extending _inference_ to longer sequence lengths than used during training. Thus, for every $n$, we indeed have different input data as you correctly mention, but we consider an _already learned fixed_ kernel at inference time.
> > > > > > >
> > > > > > > To be complete, letting $\\hat{k}(w)$ be the _fixed_ MLP and borrowing your notation that the subscript $n$ denotes sequence length, we have
> > > > > > >
> > > > > > > \\[
> > > > > > > \\hat{k}_n[m] = \\hat{k}(w)\\Big|\_{w=2\\pi m/n}.
> > > > > > > \\]
> > > > > > >
> > > > > > > In addition, you pointed out that “the value of the function at $x(n)$ from the inverse FFT is not the same as that given by the inverse discrete time Fourier transform.” This is true, but as we mentioned, our results imply that this approximation error goes to $0$ as $n \rightarrow \infty$.

---

> > > > > > > > ### Comment · Reviewer_fqtA · 2023-08-16
> > > > > > > >
> > > > > > > > Dear authors,
> > > > > > > >
> > > > > > > > I thank you for your continued discussion on this point. It seems there is a rather fundamental disagreement on the setup, and to be clear, I still do not agree with you (more on that below). Let me, however, take a step back from this particular point and return to a more holistic review of the paper after our back and forth.
> > > > > > > >
> > > > > > > > ***
> > > > > > > >
> > > > > > > > ### Stepping back
> > > > > > > >
> > > > > > > > While I'm not an expert in the NLP field, the main idea expounded by the authors appears valuable. The limitations to the potential speed-up of at best $O(\log n)$ over other methods, critiqued by some reviewers including myself, seems acceptable if there are other advantages to the technique. The preference for "flash attention" by one reviewer doesn't necessarily negate the value of this method. Rejecting papers solely because they don't achieve the state-of-the-art seems narrow. However, I defer a thorough evaluation of the experiments and practical implementations to the area chair and other specialists.
> > > > > > > >
> > > > > > > > My concerns about the theory with which I feel more suited to review still largely remain. Proposition 1 reads as self-evident and doesn't necessitate proof. It would be more fitting as a sentence rather than a formal proposition.. Theorem 1 is unclear and its bounding doesn't seem tight. There's a possibility that experiments might underscore this looseness, and if so, such experiments should be made to back up the statements. Finally, theorems 2 to 4 make a simple point, but are so couched in confusing and misleading language that the final message is missing. I still do not know why the authors are insisting on placing these statements into formal theorems. This paper is not a theory paper and heuristic justifications of the decay of the kernel would be totally fine with me (in fact, preferred since it might be understandable by the broader community).
> > > > > > > >
> > > > > > > > At the end of the day, the success of this paper does not rest on its theoretical strengths. However, papers should be correct and cleanly written so having wrong or misleading theory is a reason for not accepting the paper. All in all, I do feel that the authors should commit to significant simplifications and changes to that section for me to continue recommending acceptance.
> > > > > > > >
> > > > > > > > ***
> > > > > > > > ### Continued discussion on Theorems 2 to 4
> > > > > > > >
> > > > > > > > The setup where we fix $n$ during training and send $n \to \infty$ during inference is not one I would agree with. Nobody, in practice, ever sends the inference sequence length to infinity simply because it's just not possible. More valid assumptions are something where the inference sequence length is roughly a constant times the training sequence length or maybe even polynomial in the input sequence length. The issue I continually raise appears with both of these more realistic assumptions.

---

> > > > > > > > > ### Author Response · Authors · 2023-08-21
> > > > > > > > >
> > > > > > > > > Thank you for your comments. Based on your comments, we propose to in our next revision:
> > > > > > > > > -mention the limitations on the error bound and the asymptotics.
> > > > > > > > > -move proposition 1 to the appendix and only mention it in the main paper as a sentence.
> > > > > > > > > -fully acknowledge that the theoretical statements are a guide helping to justify the usage of the method, rather than tight theoretical results.
> > > > > > > > >
> > > > > > > > > We admit that the theory is not the primary contribution of the paper, and mainly use it to justify that the method has some theoretical basis.

---

### Official Review · Reviewer_hMrX · 2023-07-06

**Soundness:** 3 good
**Presentation:** 3 good
**Contribution:** 2 fair
**Rating:** 6
**Confidence:** 4

**Summary:**

The authors of the paper propose two modifications of a recently published alternative to attention mechanism, Toeplitz Neural Operator (TNO), which constitutes the most important part of Toeplitz Neural Networks. The application of TNO is the multiplication of the input sequence by a Toeplitz matrix. Parameters of this Toeplitz matrix are given by a lightweight feed-forward network, called Relative Position Encoder (RPE). The first proposed modification, called SKI-TNN, represents a learned Toeplitz matrix as a sum of a sparse and low-rank matrix. Thus, its multiplication by a vector has the complexity of O(nr^2 + r log r) instead of O(n log n), where r is the rank of a second summand. However, this modification can speed up only the task of bidirectional modeling. In order to speed up the causal modeling (such as autoregressive language modeling), authors view the TNO as an application of kernel to a vector. Then, they train the RPE to model the real part of the Fourier transform of this kernel. The imaginary part is then computed via the Hilbert transform of the real part. This modification does not change the asymptotic complexity, but achieves empirical speed up.

**Strengths:**

1. The article explores an important topic of speeding up the token mixing part of a general sequence modeling pipeline. Nowadays, this topic is highly relevant because of its applications in the field of NLP. The article builds off of a very recent paper [1].
2. The article creatively combines together a large body of previous work. It uses the ideas of TNN, SKI, FFT, Hilbert transform, Nyström approximation, fast causal masking, etc.
3. The experimental results show that the proposed modifications do indeed speed up the original TNN.
4. Overall, the presentation style is mathematically strict and to the point. The formulae in section 3.2.1 and in Appendix are sufficiently well-explained. Both modifications proposed in the paper are succinctly defined in Algorithm 1 and Algorithm 2. This helps the reader significantly to understand the main ideas.
5. In section 3.2.1, the authors specified not only the theoretical complexity of their modification, but also the practical limitations they meet when implementing it, and specified the practical complexity as well as theoretical.
6. The theory on the smoothness in Fourier Domain is supported with experimental visualizations
[1] Zhen Qin, Xiaodong Han, Weixuan Sun, Bowen He, Dong Li, Dongxu Li, Yuchao Dai, Lingpeng Kong, and Yiran Zhong. Toeplitz neural network for sequence modeling. In The Eleventh International Conference on Learning Representations, 2023.


**Weaknesses:**

1. The main claim of the article is the speedup achieved by the proposed modifications. The only results supporting this claim are Fig. 1 and some percentages in the text (in section 5.1). Fig. 1 shows the performance on specific tasks from the LRA benchmark. Firstly, it is not clear for which task the baseline (TNN)  is evaluated. Secondly, the choice of the tasks shown on the graph is questionable. The hardest task from the LRA benchmark, Pathfinder-X, is not shown. As for the speedups mentioned in the text: it would be better to put them all into a separate table. Moreover, it would be interesting to see a speed comparison in the form of a table similar to Table 5 from the TNN article.
2. In section 4.2, theoretical results on the choice of activations are presented. Several possible improvement ideas. The ablation study with experimental results for different activation types would be of interest. Moreover, the graphs showing the decay rate for randomly initialized networks might be improved. It would be better to leave only the lowest and highest lines and show the average line in between. Also, if you compare the rate of convergence to some baseline rate (e.g. exponential), plotting it would be appropriate. In addition, it seems to be not quite fair to compare the decay rates of trained and untrained networks.
3. While the overall presentation style is to the point, as mentioned earlier, it would help the reader if the abstract, introduction and related work were more general. Both abstract and introduction may be hard to read for an unprepared reader, as they contain too much mathematical details and not enough motivation. Moreover, some paragraphs of the introduction repeat the abstract almost word for word, while rephrasing would help the reader to understand the ideas more deeply. The related work section should describe either the ideas of mentioned papers or their connection to your paper more clearly.
4. The section 3.2.2 is a bit obscure. “Inverse time warp” is not a common term, and it is not described in the section.


**Questions:**

1. I did not understand from the paper how the training process for the sparse summand of the learned Toeplitz matrix is organized. Is it learned without RPE at all?

---

> ### Author Rebuttal · Authors · 2023-08-09
>
> Thank you for your comments and overall positive assessment, as well as your suggestions for improving the clarity.
>
> # Weaknesses
> 1. We admit that we do not have results for pathfinder-X, the most challenging dataset and task. We had difficulty achieving good performance on it for the model sizes allowed by long range arena (LRA). The original TNN paper did not have it (table 4) either for allowed model sizes, only a large model (table 14). It likely requires a larger model (at least within this class) to perform well on, violating LRA guidelines.
> 2. Thank you, these are very good points and we will add the additional experiments, including the plots for both trained and untrained networks.
> 3. We agree. In line with this, we have modified the abstract in the general rebuttal for all authors. We will also revise the introduction and related work. In the introduction in particular, in the 2nd paragraph we will add a bit summarizing the computational bottlenecks in practice and not only mathematically. In the third paragraph, we will focus on the high level ideas rather than aiming to translate the equations into words.
> 4. Thank you for the feedback. The inverse time warp is the strictly decreasing function $x(t)$ that allows us to map the interval $[0, \infty)$ to the interval $(0, 1]$. We use this so that we do not have to extrapolate beyond our inducing locations. We will add this additional explanation and explicitly say “We use an inverse time warp $x(t)$ and let…” in section 3.2.2.
>
>
> # Questions
> 1. Indeed, no MLP RPEs are used to represent the sparse component of the decomposed Toeplitz matrix (and in fact no MLP RPEs are used in SKI-TNN at all). We describe in section 3.2 that “Applying the action $T_{\textrm{sparse}}x$ of $T_{\textrm{sparse}}\in R^{n\times n}$ with $m$ non-zero diagonals is equivalent to applying a 1D convolution layer with filter size $m$.” It is thus learned as the parameters of the separate 1D convolution layer.

---

> > ### Comment · Reviewer_hMrX · 2023-08-14
> >
> > Thank you for addressing my concerns!

---

> > > ### Author Response · Authors · 2023-08-22
> > >
> > > Thank you, we are pleased that we were able to do so!

---

### Official Review · Reviewer_aAC6 · 2023-07-12

**Soundness:** 2 fair
**Presentation:** 2 fair
**Contribution:** 2 fair
**Rating:** 4
**Confidence:** 4

**Summary:**

The paper presents several techniques to speed up Toeplitz neural networks (TNNs). In particular, TNNs use convolution of length n (sequence length of the input) and so scales as O(n log n), and TNNs have many calls to the MLP that generate relative positional encoding (RPE) and decay bias. To reduce the time of convolution, for bi-directional modeling the paper proposes to approximate the Toeplitz matrix as a sum of a short convolution and a low-rank matrix, which results in O(n + r log r) complexity where r is the rank of the approximation. For uni-directional model (e.g. auto-regressive modeling), the paper proposes to parameterize the convolution directly in frequency domain and uses the Hilbert transform to obtain the imaginary part from the real part of the filter to ensure causality. The approximation error is then analyzed. Validation on language model (Wikitext-103) and long-range benchmark (LRA) show that the approximation lead to some speedup (10-15%) and the quality stays around the same.

**Strengths:**

1. The idea of using asymmetric Nystrom to approximate the Toeplitz matrix is quite clever. This allows a decomposition into a sparse and a low-rank component, which leads to asymptotically faster algorithm in the case of bi-directional modeling.

2. While uni-directional modeling prevents the Nystrom technique due to causal masking, parameterizing the filters directly in the frequency domain is able to overcome this challenge. While this is not asymptotically faster, it avoids one inverse FFT per layer and leads to some speedup.

**Weaknesses:**

1. Unclear what problem the paper is trying to address, and how it is motivated.
The intro starts out with Toeplitz neural networks, and the paper aims to make it faster. However, it's not clear why we want to make these faster, and what we would enable if we make these faster. Are they being used in very large-scale tasks? Are they being scaled to very long sequences?
While the technical contributions are solid, it's not clear to me why the paper chose to tackle this problem.

2. Unclear what the technical challenges are.
- The paper mention that they want to avoid O(n log n) computation. But in practice O(n log n) isn't very slow, especially on GPUs. FFTs are pretty much bounded by memory bandwidth, and they take only 2-3 times as long as any pointwise operation. So if the goal is to speed up TNNs, then it makes more sense to have an efficient implementation, rather that using algorithms that faster asymptotically (O(n + r log r)) but is slower than a hardware-friendly algorithm (line 150, where using matmul with O(n r^2 + r log r) is faster).
- The paper mentioned "many calls to the RPE". Why is this a problem? Showing a profile of how much each operation is taking will be much more convincing. That would motivate the approaches in the paper much better.
Without knowing how long each operations in TNNs are taking, how do we know that we're solving the right problem?

3. Lack of detailed speed benchmark. Given the goal is to speed up TNNs, I would have expected one of the main results to be speed benchmarks, across different sequence lengths, on different devices, to show the tradeoff. In the main paper, speed is only reported in Figure 1b, which is end-to-end speed for a particular sequence length (512).
How do we know that we're close to the maximum speed on these devices (GPU)? Or are we still far from optimal? When we speed up convolution and RPE, what is the remaining bottlenecks.
Having these would make the paper stronger.

At sequence length 512, an optimized implementation of attention (e.g. FlashAttention) is likely faster than FFT and the method in this paper. This is my impression as the Hyena paper [1] reports that TNNs are not faster than FlashAttention until sequence length > 4k.

[1] Hyena Hierarchy: Towards Larger Convolutional Language Models. Poli et al. 2023.

**Questions:**

3.2.2 I don't understand what's an inverse time warp. There doesn't seem to be an explanation of what that is in this section.

**Limitations:**

Not necessary.

---

> ### Author Rebuttal · Authors · 2023-08-08
>
> Thank you for your comments, which have helped us create plots that we believe will strengthen the paper. We answer your questions below, and hope they are adequate to consider altering your assessment.
>
> # Strengths
>
> 1. Thank you. Note that the relationship is actually reversed: it is the decomposition into a sparse and low-rank component that allows the use of asymmetric Nystrom, leading to the fast algorithm.
>
> # Weaknesses
> 1. Good point. We have rewritten the abstract to reflect this. We aim to help train language models more efficiently (which has a direct dollar impact on training cost) and to help extend to longer contexts. TNNs are a recent model successfully achieving both, and thus we aim to continue to improve upon their success.
> 2.
> * We agree with the premise and also observed the importance of efficient implementations over lower theoretical complexity. However, this depends on the value of n. We demonstrate a substantial empirical speedup against TNNs right now across a range of sequence lengths and tasks, but the asymptotic speedup we achieve is also valuable to maintain the possibility for further speedup in the future as sequence lengths seem to be increasing.
> * We have added some profiling to the pdf attached to the general rebuttal. We show that for the TNO, the RPE takes approximately 2/3 of the time of the primary repeated pattern. Upon closer inspection (which there was no space to include the zooming in of), much of this is due to poor GPU utilization between layers. We find that our SKI-TNN approach improves GPU utilization by approximately 45% at sequence length 512.
> 3.  We have provided in the attachment an initial benchmarking on an A100 for SKI-TNN for 512 and 2048 sequence lengths. We will add this for more sequence lengths and 1-2 more devices in the next revision. We see speedups (how much the wall clock time is reduced) of 25% and 30%, respectively. We also see GPU memory reductions of 17% and 42%, respectively. Note that because we are proposing an approximation rather than a purely algorithmic speedup for the same computation, speed benchmarks alone still leave the question open on whether performance is not also adversely impacted. However, training all approaches to completion would be prohibitive in terms of compute costs. Thus, we chose a selected set of tasks that we hoped would cover a reasonable sample of representative settings. One of our main results in Fig 1a does in fact provide speed comparisons in this way on LRA, a commonly used benchmark. Nonetheless, we will provide some more speed benchmarking, noting the nuance needed to interpret the results of the comparison due to approximation.
>
> Note that the Hyena paper and TNNs treat two distinct models, and the Hyena paper does not run any experiments using TNN’s. Their results are using the Hyena module to replace the self-attention module within a Transformer, which is an altogether distinct architecture from the TNN, although both use the action of Toeplitz matrices. In fact, TNNs are faster than Flash starting at least from sequence length 1k (table 5 in the TNN paper). Further, TNNs are much faster and have much higher score on long range arena (LRA). See figure 1 in the TNN paper. Our methods are much faster still than TNNs on 1d tasks, with only a small sacrifice on score in the SKI case and sometimes an improvement in the FD case (we would still have a much higher score than Flash attention). Finally, Our FD-TNN approach achieves lower perplexity than flash attention when doing pre-training (table 1 in our paper) at sequence length 512.
>
> # Questions
> Thank you for the feedback. The inverse time warp is the strictly decreasing function $x(t)$ that allows us to map the interval $[0, \infty)$ to the interval $(0, 1]$. We will add this additional explanation and explicitly say “We use an inverse time warp $x(t)$ and let…” in section 3.2.2.

---

> > ### Comment · Reviewer_aAC6 · 2023-08-22
> >
> > Thanks to the authors for the explanation. I have increased my rating.
> >
> > However, one of the the motivations for improving complexity from O(n log n) to O(n) remains not very convincing to me. This is asymptotic and depends on the value of n, but in practice for most values of n the difference between FFT and a linear time operation (e.g elementwise) is quite small. I have just benchmarked (on A100 GPU) that FFT takes at most 4x time the time of an elementwise operation such as ReLU, up to n of size 16M. Sequence length for language modeling is no where near 16M. My point is, for most values of n that we can about (e.g., up to 16M), we can think of FFT taking 4n time and elementwise function taking n time.
> >
> > From the profiling result it seems RPE (of time O(n)) might be important to improve. However, profiling seems to suggest that the time is being spent on CPU overhead and not GPU operation, so one worthwhile direction is just to implement these efficiently (e.g. writing in a lower-level language instead of calling PyTorch, or using torch.compile or CUDA graph).

---

### Official Review · Reviewer_qNcu · 2023-07-28

**Soundness:** 3 good
**Presentation:** 3 good
**Contribution:** 2 fair
**Rating:** 5
**Confidence:** 3

**Summary:**

The paper proposes to reduce the computational complexity of Toeplitz neural networks. TNNs are a new form of network for sequence modeling that reduces space complexity of the attention matrix to allow for longer sequences.TNN model consists of a stack of Gated Toeplitz Units (GTU) that includes TNO (Toeplitz Neural Operator) that does token mixing with relative positioning. Then, GTU is a modified GLU layer injected with the proposed Toeplitz Neural Operator (TNO).
The paper addresses the TNNs efficiency limitations: 1) super-linear computational complexity 2) many calls to the RPE: for each layer, one call per relative position. Thus, the paper proposes to reduce both the complexity of sequence modeling and of the relative positional encoder. The work proposes solutions by means of both the Structured Kernel Interpolation (SKI) [2] and working with frequency domains.
It does so through:
–	Approximating Toeplitz matrix using low-rank approximation and replacing the RPE MLP with linear interpolation and using Structured Kernel Interpolation.
(for O(n) complexity, that is use linear interpolation over a small set of inducing points to avoid the MLP entirely   -using an inverse time warp to handle extrapolation to time points not observed during training)
–	Causal training, SKI does not bring benefits, so instead they eliminate explicit decay bias by working in the frequency domain, using Hilbert transform (to force causality) and also use some smoothness
–	 For the bidirectional case, they eliminate the FFT applied to the kernels.


**Strengths:**

The work includes a number of solutions to improve TNN speed-up (addressing RPE MLP, the FFT, and the decay bias).

RPE is a neural network to obtain relative position embedding to obtain entries in Toeplitz matrices. These entries could be evaluated with stationary non-SPD kernel which is a good idea.

So first decomposing Toeplitz matrix and then using interpolation for MLP is a comprehensive pipeline.

The theory part makes the arguments more sound, and the explanations in supplementary materials are fairly abundant.

The experiments on LRA show good predictive performance on long range data and on wikitext some speed-ups.


**Weaknesses:**

The major paper of the paper talks about the SKI to accelerate the TNNs but in experiments SKI is only shown in the LRA experiment and does worse than both TNN and FD-TNN

As mentioned in the paper, doing sparse-dense multiplication in practice can be slower than dense-dense matrix multiplication (but that is only part of the potential speed-up)


**Questions:**

Can you elaborate on “We can interpret Tlij as evaluating a stationary non-SPD kernel kl (i j) = |i j| RPEl (i j). Thus Tl can be interpreted as a pseudo or generalized Gram matrix”

What is the practical length of a sequence where the benefits are seen over TNN?

A suggestion: some more elaboration in the main paper on causality aspect would be helpful, what is fast causal masking, what [33] says about ensuring causality instead of citing it.

Minor: Provide better notation explanation, e.g. what is PPL, architectures, ListOps, Pathfinder, etc.


**Limitations:**

Yes.

---

> ### Author Rebuttal · Authors · 2023-08-09
>
> Thank you for your comments. We address them based on the relevant section.
>
> # Weaknesses
>
> * While SKI-TNN has slightly worse predictive performance than TNN, in most cases (2/3 of the LRA cases and the bidirectional pre-training), the degradation is minimal, while the speed improvements are dramatic. The ability to make this trade-off between model quality and speed is an important contribution of SKI-TNN over TNN and FD-TNN. Regarding pre-training results on Wikitext 103 for SKI-TNN, we now have these results in the main attached pdf and will include them in the Appendix, demonstrating similar increases in speed trading off a small quality degradation. Finally, we would consider *both* SKI-TNN and FD-TNN to be core contributions of this paper, and not only SKI-TNN.
>
> # Questions
>
> * A Gram matrix is a matrix of inner products. Evaluating symmetric positive definite (SPD) kernels corresponds (Mercer's theorem) to an inner product. Thus, a matrix of pairwise kernel evaluations for a set of observations is a Gram matrix. In our case, instead of evaluating such a matrix for an SPD kernel, we are evaluating it for a non-SPD kernel. Thus we consider it to be _like_ a Gram matrix and call it a generalized Gram matrix.
> * Regarding the practical sequence lengths where we see benefits over TNN, our experiments measuring both speed and quality together are limited to N=512 for causal and bidirectional pretraining, and the range of lengths for the LRA tasks (between 1024 and 4096).  For pre-training on Wikitext-103, we anticipate equivalent quality between FD-TNN and TNN (or similar quality degradation between SKI-TNN and TNN) at longer sequence lengths. We have also added a figure in the general pdf attachment showing at sequence lengths 512 and 2048, pre-training using SKI-TNN is faster and uses less memory than using TNNs.
> * We will add this. We have some analysis in Appendix B, but we will add some comments in the main paper.
> * We will add these descriptions in an appendix, which we will cite in the main paper.

---

### Author Rebuttal · Authors · 2023-08-09

We thank the reviewers for their helpful critiques and comments, including “The article creatively combines together a large body of previous work” and “The current paper presents a nice idea.” There were two areas where multiple reviewers had critiques, and we address them here. The first is that several additional experiments were requested, including RPE profiling, sequence length profiling and SKI language model pre-training. We have included some results for each of them in the attached pdf. For RPE profiling, visually it looks like it takes about 2/3 of the Toeplitz Neural Operator (TNO) time. For sequence length pre-training profiling for SKI-TNN we show 512 vs 2048 for speed, memory usage and GPU utilization. At sequence length 512 our approach has a 25% reduction in time per step, requires 17% less GPU memory, and has an approximately 45% increase in GPU utilization. We see that at sequence length 2048 we have 30% reduction in time per step and requires 42% less memory, while the GPU utilization gap becomes small. Due to an almost complete unavailability of A100s on both GCP and Lambda labs during this rebuttal period we were unable to profile further sequence lengths, but will add more for the next revision. We also do SKI language model bidirectional pre-training, and find that over 50k steps we reach nearly the same validation perplexity with SKI-TNN as with TNN. We will add further profiling for FD-TNN in the next revision.

The second area of comments relates to the fact that the abstract and introduction may have had too little motivation for the general problem and focused too much on the technical solution approach. We propose to modify the abstract as follows, and will modify the introduction and related work accordingly as well.

An important language modeling challenge is handling long context. A key is alleviating computational bottlenecks, such as the quadratic complexity of self-attention, while maintaining expressivity. Toeplitz Neural Networks (TNNs) are a recent impressive language model aimed at long context. They have two core computational bottlenecks: 1) an $O(n\log n)$ FFT and 2) an $O(n)$ relative positional encoder (RPE) multi-layer perceptron (MLP). In this paper, we explore whether we can reduce the number and size of the FFTs, and whether we can eliminate relative positional encoder (RPE) multi-layer perceptrons (MLPs). We consider two approaches trading off complexity vs quality. In the first, we note that the RPE is an asymmetric indefinite kernel and the Toeplitz matrices are pseudo-Gram matrices, and that learned kernels exhibit structure motivating an approximate sparse and low rank decomposition. We do a small 1d convolution for the sparse component. For the low rank component, we replace the RPE MLP with linear interpolation and use Structured Kernel Interpolation (SKI) for $O(n)$ complexity, enabling use of a smaller FFT and Toeplitz matrix with a small loss in quality. For our second approach, we note that by working in the frequency domain, we can avoid an explicit decay bias. In the bidirectional setting, this needs one fewer FFT and is straightforward to implement. In the causal setting, we enforce causality by representing the real part of the kernel’s frequency response with the RPE and computing the imaginary part via a Hilbert transform. These frequency approaches maintain $O(n \log n)$ complexity but achieve an absolute speedup with no observed loss in quality. We improve on speed and sometimes score on the Long Range Arena (LRA).

---

### Decision · Program_Chairs · 2023-09-21

**Decision:**

Reject

**Comment:**

The reviewers generally agreed that this paper presents interesting results on speeding up Toeplitz neural networks. However, several concerns ultimately led to a decision to reject. We encourage the authors to address these concerns as they revise the paper. The main concerns discussed were:

- The paper needs significant improvement in its presentation and problem motivation. Currently, the paper is very difficult to follow for a non-expert on the immediately adjacent work. We appreciate the authors' commitment to improving this aspect of the paper (including e.g., suggesting a new abstract) but feel that the needed changes are significant enough that further review after changes would be valuable.
- While the theoretical results are primarily meant to complement the empirical results, and may not be the main contribution of the paper themselves, they need to be technically correct and sound if presented as formal results. Reviewers were concerned about the formality of the theoretical results and in particular clarity over exactly what setting they apply to. We encourage the authors to address these concerns.
- After discussion, we felt that the motivation of achieving O(n) complexity as compared to O(n log n) needs to be strengthened, especially given that: 1) the O(n log n) complexity is due simply to an FFT, which in practice is extremely efficient and 2) as pointed out by one reviewer, the O(n) complexity hides many problem dependent factors, which are not clearly bounded, and in fact, which in theory seem to have to grow with order at least log n, negating the improved asymptotic complexity.